



# Water-balance and hydrology research in a mountainous permafrost watershed in upland streams of the Kolyma River, Russia: a database from the Kolyma Water-Balance Station, 1948-1997

Olga Makarieva[1,2,3], Nataliia Nesterova[3,4], Lyudmila Lebedeva[2], Sergey Sushansky[5,*]

[1]*Gidrotehproekt Ltd, St. Petersburg, 199178, Russia*
[2]*Melnikov Permafrost Institute, Merzlotnaya St., 36, Yakutsk 677010, Russia*
[3]*Saint Petersburg State University, Institute of Earth Sciences, 7/9 Universitetskaya nab, St. Petersburg, Russia 199034*
[4]*State Hydrological Institute, Department of Experimental Hydrology and Mathematical Modelling of Hydrological Processes, 23 2-ya liniya VO, St. Petersburg, Russia 199053*
[5]*formerly at: the Kolyma water-balance station, Stokovoye, Magadan district, Russia*
*\*retired*

Contact author: omakarieva@gmail.com

**Abstract:** As of 2017, 70 years have passed since the beginning of work at the Kolyma water-balance station (KWBS), a unique scientific research hydrological and permafrost catchment. The volume and duration (50 continuous years) of hydrometeorological standard and experimental data, characterizing the natural conditions and processes occurring in mountainous permafrost conditions, significantly exceeds any counterparts elsewhere in the world. The data are representative of the vast territory of the North-East of Russia. In 1997, the station was terminated, thereby leaving Russia without operating research watersheds in the permafrost zone. This paper describes the dataset containing the series of daily runoff from 10 watersheds with area from 0.27 to 21.3 km$^2$, precipitation, meteorological observations, evaporation from soil and snow, snow surveys, soil thaw and freeze depths, and soil temperature for the period 1948-1997. It also highlights the main historical stages of the station's existence, its work and scientific significance, and outlines the prospects for its future, where the Kolyma water-balance station could be restored to the status of a scientific research watershed and become a valuable international center for hydrological research in permafrost. The data is available at https://doi.pangaea.de/10.1594/PANGAEA.881731.

**Keywords:** water-balance and hydrological research, continuous permafrost, Kolyma River, Kolyma water-balance station, streamflow, thaw/freeze depth, precipitation, snow cover, evaporation from soil and snow



## 1. Introduction

In 2017 we celebrate 70 years since work on organizing the Kolyma Water-Balance station (KWBS) began. This hydrological and permafrost research catchment has accumulated standard and experimental data unique both in terms of their amount and duration.

In the paper «Save northern high-latitude catchments» Laudon et al. (2017) recognize the KWBS as a currently functioning scientific station, even though scientific research was suspended here 20 years ago, and nowadays only standard observations at the meteorological site and the runoff gauge are carried out.

Eurasia contributes 75 % of the total terrestrial runoff to the Arctic Ocean and three of the four major Arctic rivers are located in Siberia (Shiklomanov et al. 2002). Peterson et al. (2002) suggested that the net discharge from the six largest Eurasian rivers flowing into the Arctic Ocean (Severnaya Dvina, Pechora, Ob', Yenisey, Lena, and Kolyma) increased by 7% during the 20th century. As it is also mentioned by Laudon et al. (2017), the number of scientific and hydrological research stations in the Northern regions of the world has decreased by 40%, and it happened alongside the most significant climate change in the Arctic in recorded history.

The Kolyma Water Balance Station (KWBS) is located in the headwaters of the Kolyma River, in a mountainous region of continuous permafrost (Fig. 1). Runoff formation conditions at the station are representative for an immense territory of the upper Kolyma River basin and mountainous regions of North-East Russia. Although there are large mountainous areas in other cold regions of the world in Canada, the USA and Europe, the combination of extremely severe climate and continuous permafrost creates unique conditions in North-East Russia.

To the best of our knowledge, the first systematic cold-region hydrology observations in North America begin not earlier than the 1960s. The Caribou-Poker Creeks Research Watershed was established in 1969 (Hinzman et al., 2002).

At the KWBS there were 10 hydrological gauges at catchments ranging between 0.27 and 21.3 km$^2$, two meteorological plots, 55 (in total) precipitation gauges, over 30 frost tubes (cryopedometers), several groundwater wells, evaporation, water-balance and runoff plots. In addition, regular snow surveys were conducted, as well as experimental investigations of specific hydrological and permafrost processes.

During the period 1948 to 1997, the KWBS accumulated a huge amount of data on hydro-meteorological and special observations of a unique duration (40-50 years) that characterize the natural setting, which, on the one hand, are nearly ungauged, and on the other hand, are representative of the vast territory of North-East Russia and to some extent of other alpine cold-climate areas in Europe and North America. The data were published in 40 reports, the first one covering the period 1948-1957. Following issues were published annually (KWBS Observation's materials, 1959-1997).

After 1997, water balance observations at the KWBS were ceased. One weather station and five runoff gauges functioned at the KWBS up to mid-June, 2013, when an extreme flash flood destroyed four level gauges. Nowadays only standard observations are conducted at the Nizhnyaya meteorological site and at the Kontaktovy (Nizhny) runoff gauge.

Observation results were reflected in more than 250 publications, dedicated to different aspects of runoff formation in the continuous permafrost region, active layer dynamics, underlying surface structure and its influence on hydrological processes. Based on the KWBS materials, the following research was carried out:

- water balance formation (Boyarintsev, 1980; Boyarintsev, Gopchenko, 1992; Kuznetsov et al., 1969; Suchansky, 2002; Zhuravin, 2004; Lebedeva et al., 2017),
- peak and spring flood runoff in small rivers in the permafrost zone (Boyarintsev, 1988),
- base flow (Boyarintsev, Nikolaev, 1986; Glotov, 2002),
- principles of runoff cryo-regulation (Alekseev et al., 2011),
- ice content dynamics of rocky talus deposits (Bantsekina, 2001),
- processes of intra-ground condensation (Reinyuk, 1959; Bantsekina et al., 2009; Boyarintsev et al., 1991),



•   floodplain taliks in continuous permafrost (Mikhaylov, 2013) and many others.
Collected data were also used for development and testing different geoscience models of:
•   runoff formation (Gusev et al., 2006; Kuchment et al., 2000; Vinogradov et al., 2015;
Lebedeva et al., 2015; Semenova et al., 2013),
•   climatic aspects (Shmakin, 1998),
•   land surface and vegetation dynamics (Tikhmenev, 2008).
In this paper, we present a hydrometeorological and permafrost related dataset for 50
years from 1948 – 1997 for the Kolyma Water-Balance Station (KWBS), the Kontaktovy Creek
watershed, which is representative for vast mountainous territories of continuous permafrost
zone of Eastern Siberia and the North-East of Russia. This dataset is unique in terms of its
volume and duration of hydrometeorological standard and experimental data. It may be used in
many research tasks, but is of particular importance in studying runoff formation processes and
model development in permafrost regions.

## 2. Site description

The Kolyma water-balance station is located in the Tenkinsky district of the Magadan
region of Russia within the Upper-Kolyma highland. The station's territory – the Kontaktovy
Creek catchment with area 21.3 km$^2$ – is a part of the Pravy Itrikan River which flows into the
Kulu River basin, which is the right tributary of the Kolyma River. The station is located 16 km
from the Kulu village settlement. It is characterized by a mountain landscape, typical for the
upper reaches of the Kolyma River. The territory of the basin is severely cut up with creek
valleys. These valleys are narrow, with steep slopes, and watershed lines are mostly well
delineated. Absolute elevations of the basin range between 823 m a.s.l. near the Kontaktovy
Creek outlet and 1700 m a.s.l. at watershed divides. The length of the creek is 8.9 km. The
catchment is extended in the latitudinal direction and has an asymmetric shape. The slopes of the
catchment area have mainly southern exposure (53% of the slope area), the slopes of the
northern and eastern exposure have a 24% share, the western – 23%. The density of river
network in the basin is 2.5 km per square km. The main river canal is meandering. The steepness
of the slopes ranges from 200 to 800‰ (Fig. 1).
The station is located in the continuous permafrost zone. Permafrost thickness varies
from 120 to 210 m in valleys and can reach 300-400 m in highlands, following the relief.
Seasonal soil thaw depth depends on slope exposition, altitude and landscape and changes from
0.2-0.8 m on north-facing slopes to 1.5-3.0 m on south-facing ones.
KWBS is situated in the transitional zone between forest-tundra and coniferous taiga. Soil
types vary from stony-rock debris to clayey podzol with partially decayed organic material
underlain by frozen soil and bedrock. Most of the KWBS area is covered by rocky talus,
practically without vegetation (34%). Dwarf cedar and alder shrubs are common at south-facing
slopes and cover about 27% of the territory. Larch sparse woodland with moss-lichen cover is
typical for steep north-facing slopes (12%). Open terrain larch wood (15%) and swampy sparse
growth forest with minimal permafrost thaw depth, constant excessive stagnant moisturizing,
tussock or knobby microrelief (12%) characterize creek valleys. The estimates of landscape
distribution are given here after Korolev (1984).

### 2.1 History of KWBS

The Kolyma water balance station (KWBS) was established on October 15, 1947 and
was initially known as the Itrikanskaya runoff station of the Dalstroy (Far North Construction
Trust organized in 1931) Hydrometeorological Service. In 1948-1956 and 1957-1969 it was
called the Kulinskaya and the Kolyma runoff station respectively. The primary goal of this
station was studying runoff formation processes in small river catchments in mountain
permafrost landscapes, typical for northeast USSR.
As soon as May, 1948, the first runoff observations at the Kontaktovy Creeks and
Vstrecha brook were launched, as well as regular observations at the Nizhnyaya weather station



(850 m a.s.l.). A few months later, on September 1, 1948, observations at the Verkhnyaya
weather station (1220 m a.s.l.) were started. In 1948, stage gauges Sredny, Nizhny and Vstrecha
were equipped with automatic water level recorders, gauging footbridges and flumes.
During the period 1949-1957, at the Vstrecha brook catchment, a rain-gauge network was
organized. Runoff gauges at the Severny, Dozhdemerny, Vstrecha brooks were equipped with
various hydrometric facilities. Observations on soil, water and snow evaporation, soil freezing
and thawing commenced, as well as experimental observations at a runoff plot.
At the end of the 1940s and early 1950s, technical staff of the station were mainly former
convicts. During the first few years, the workers of the station built houses for themselves,
collected firewood and organized the household. The winter of 1955-56 appeared to be
especially severe for the staff, since due to the deep snow cover it was difficult to move around
the territory of KWBS, there was no transport connection with the Tenkinskaya highway,
delivering of firewood, needed for heating houses and service buildings, was also difficult. When
it was impossible to get to the highway by car, bread and mail were delivered from the Kulu
village settlement utilizing horses once every 7-10 days.
Twenty to twenty five staff members were accommodated in three small huts, hardly
suitable for living. That winter they mainly had to collect and prepare firewood in the afternoon;
in the morning everybody had to go (despite their rank or position) in deep snow and at -50°C to
the nearest small river valley looking for firewood, then they pulled it back home, where they
were firing furnaces. Only by the time it got dark, it became warm enough to stay in the work-
room and they could start observation data processing.
The working day lasted till 10 or 11 p.m. Since there was no electricity, they used
kerosene lamps filled with a mixture of petrol and salt. In summer 1956 there were only 13
people left at the station, some of them were taken to help with haymaking to prepare hay for
their subsidiary holding that consisted of two cows and a horse (as recollected by the Chief of the
station V.G. Osipov, the hydrologist-technician A.I. Ipatieva, Informational letter…, 1988).
In 1957 the station was handed over to the jurisdiction of the Kolyma Hydro-
meteorological service administration, and in 1958 it was partially connected to electricity. At
that time there were active steps taken toward fitting out the station with new types of devices
and equipment, engaging new specialists in hydro-meteorology, and building accommodation
facilities.
In 1960 runoff observations at the Yuzhny brook were begun, rationalization of the
precipitation network was continued, and radio rain gauges were installed.
In 1963 two new water-balance sites (##2 and 3) were organized.
In 1968 runoff measurements were started at the unique research object, at the Morozova
brook catchment, which has no vegetation cover and is composed of rocky talus.
In 1969 the Kolyma runoff station was renamed into the Kolyma Water Balance station
(KWBS). In these years there was a transition to broad experimental water balance observations
of all of its elements and to an enhanced technical level of research.
Since 1970, the KWBS carried out snowpack observations at avalanche sites of the
Tenkinskaya road, as well as stratigraphy, temperature and physical and mechanical properties of
snow at four sites. Since 1980 there were introduced additional observations on dynamics of
icing formation at the Kontaktovy creek. In 1982 observations on soil moisture were started at 3
agro-hydrological sites at the fields of the «Kulu» state farm.
In 1976 the station hosted a delegation of USA scientists. They highly praised the
professional and personal qualities of the station's staff members, their commitment, on which
extensive field studies and theoretical works were based, despite the equipment being rather
simple and living conditions extreme. According to Slaughter and Bilello (1977), the data
recorded at the KWBS, were unique and unprecedented for world practice.
Since the beginning of the 1990s, the research program at the station has been gradually
cutting back. After 1997, water balance observations at the KWBS were ceased. One weather
station and five runoff gauges functioned at the KWBS up to mid of June, 2013, when an



extreme flash flood destroyed four level gauges. Nowadays only standard observations are
conducted at the Nizhnyaya meteorological site and at the Kontaktovy (Nizhny) runoff gauge.

### 3. Data description
#### 3.1 Meteorological observations

The observations of meteorological elements were carried out at three meteorological
stations in different periods (Fig. 2). The database includes daily values of air temperature, water
vapour pressure, vapour pressure deficit, atmospheric pressure, wind speed, low and total cloud
amount, and surface temperature (Table 1).
The meteorological station Verkhnyaya (1220 m a.s.l., 1948-1972) was located in the
upper reaches of the Dozhemerny brook in the saddle between two hills. The horizon is closed
by the hills from the south and north which are at 30-40 m distance from the site. The horizon is
open from the east and west, strong winds are observed here. The surface at the station plot is
hummocky, covered with grassy vegetation with no woody vegetation around it. The nearest
building – the station house – was located 48 m away from the station plot. The depth of
seasonal thaw of permafrost reaches 1.5 m.
The meteorological station Nizhnyaya (850 m a.s.l., 1948-1997) is located on the edge of
a larch forest, on the terrace-watershed between the Kontaktovy creek and the Ugroza brook,
which has a slight slope to the SW. The nearest trees are located 50 m away, the buildings – 100
m from the station. The site is surrounded by mountains up to 1400 m a.s.l., the nearest of them
are at a distance of 200-500 m. The angle of the horizon is 11 degrees. The height of the weather
vane is 11.3 m. The surface at the station plot is covered by hummocks, with moss, peat and
individual bilberry and blueberry bushes. The area is surrounded by a sparse larch forest from
the north, east and south. The depth of permafrost seasonal thaw reaches 1.5 m.
The meteorological station Kulu (670 m a.s.l., 1981-1991) was located on the right cliff
of the broad valley of the Kulu river. The slope has a western exposure (4-6 degrees). The angle
of the horizon is 4 degrees. The height of the weather vane is 10.7 m. The area is surrounded by
larch trees of 6-8 m height from the west, north and east. The soils are loamy with the inclusion
of small gravel. The underlying surface consists of berry, grass and sphagnum mosses,
sometimes bare soil. The depth of seasonal thawing of permafrost reaches 1.8-2 m.
In September 1992, the Kulu station (635 m a.s.l., 1992-1997) was moved to the
residential building of the KWBS at the south-eastern part of the Kulu village. Residential and
technical buildings are located around the meteorological plot. There was a road to the south of
the station. The soil is marshy, covered with rubble and grass. The angle of the horizon is 4
degrees. This new location of the Kulu station is marked as Kulu2 station in the database.
The observations at all meteorological plots were conducted according to the program of
a meteorological station of the 2nd category.
The climate of the study area is severely continental with harsh long winters and short but
warm summers. Average annual temperature at the Nizhnyaya meteorological plot during 1949-
1996 is -11.3 °C. Mean monthly temperature in January was -33.6 °C, in July +13.2 °C (Fig. 3-
4). The absolute minimum daily temperature of -53.0 °C was registered in 1982 and the absolute
maximum daily temperature was +22.8 °C (1988). The period of negative air temperatures lasts
from October to April, freeze-free period is, on average, 130 days long.
Air temperature inversions are observed at the KWBS. In December air temperature
gradient reaches +2.0, in May it accounts for -0.5°C (100 m)$^{-1}$ respectively. Mean wind velocity
is more than twice higher at Verkhnyaya station in comparison with Nizhnyaya station (3.0 and
1.3 m s$^{-1}$ accordingly).
#### 3.2 Precipitation

In total, the precipitation was observed at 47 gauges within KWBS territory during
different periods. Continuous daily all-year around precipitation data is available for the period
1948-1997 for the gauge (#12) at meteorological station Nizhnyaya and for the gauge (#54) at



meteorological station Kulu for 1981-1997. Four gauges have the data of daily totals during
warm season for the period for more than 30 years and another 18 gauges for different shorter
periods. Usually the start of daily observations at those gauges was initiated by the beginning of
snowmelt period and lasted until the end of September. Monthly sums of precipitation were
measured at 30 gauges, 10-days and 5-days sums – at 21 and 18 gauges respectively.
Rain-gauge stations for measuring daily precipitation totals in 1948 were equipped with
the Nipher-shielded rain gauges and later – with the Tretyakov-shielded precipitation gauges. In
1948-1958 the observations were carried out with both devices in parallel, after 1959 only
Tretyakov gauges were used.
There were three main types of precipitation gauges which were used at the KWBS – the
Tretyakov, the Kosarev and pit rain gauge (Fig.5). In 1960-1963 there was an attempt to register
precipitation with automatic radio-precipitation gauges, but due to improper performance of the
devices those observations were stopped.
In 1988, precipitation observations at the KWBS were carried out with 36 precipitation
and rain gauges, distributed relatively evenly throughout the area and altitudinal zones. Average
density of the precipitation network at that time accounted for 1.6 units per 1 $km^2$.
Precipitation at Nizhnyaya meteorological plot in the 1949-1997 period varied from 229
(1991) to 474 (1990) mm per year with mean value of 342 mm. Maximum and minimum amount
of precipitation is observed in July and March and account for 71 and 7 mm respectively (Fig. 4-
261 5).
### 3.3 Snow surveys
Snow cover at KWBS is formed in the first weeks of October, and melts in the third week
of May. Snow cover observations were started in 1950 and initially conducted at two
catchments, at two meteorological plots and four typical squares. In 1959-1960 the number of
catchments with snow surveys reached five. Up to 1971, snow surveys were conducted once per
month starting in November and finishing in May at small catchments (the Severny, Yuzhny,
Dogdemerny and Vstrecha) and once before spring snowmelt at the Kontaktovy – Nizhny. Since
1972, the observations were reduced to one survey per year (usually at the end of April) for all
watersheds. Table 2 shows the number of snow routes, their total length and number of
measurement points, including their distribution among different landscapes of the catchments
(Fig. 6). Snow depth was measured every 10 m, snow density – every 50 m.
Mean, maximum and minimum observed snow water equivalent (SWE) before spring
freshet at the Kontaktovy – Nizhny accounted for 121, 213 (1985) and 59 (1964) mm
respectively in the 1960-1997 period. In general, rocky talus and tundra bush landscape are
characterized by lower SWE due to wind blowing. Much snow is accumulated in the forest
landscape. However, at the Morozova brook watershed which is fully covered by rocky talus
landscape, mean SWE before snowmelt was estimated as 161 mm with the maximum value of
298 mm observed in 1985 reaching 0.99 m snow height (Table 3, Table 4, Fig. 7).
### 3.4 Soil evaporation
From 1950 to 1953, five soil evaporation plots were opened at the sites with diverse
underlying surfaces and different expositions and altitudes. The plots were equipped with the
Rykachev and Gorshenin evaporimeters with evaporation area 1000 $cm^2$. The observations until
1958 are considered to be approximate due to the absence of accompanying rain gauges and
scales of required accuracy. From 1958 to 1966, the measurements were conducted at the soil
evaporation plot, located near the Nizhnyaya weather station. The observations of evaporation
were carried out with two evaporimeters GGI-500 and the Rykachev evaporimeter, precipitation
– with a rain pit-gauge. Three soil evaporation sites were established in different landscapes – in
the Vstrecha (1967), Morozova (1971) and Yuzhny (1977) brooks basins. The measurements
were carried out with standard weighing evaporimeters GGI-500-50, which, due to the physical



proximity of permafrost, were changed to GGI-500-30, meaning that their height was decreased
to 30 cm.
Average values of annual soil evaporation were previously estimated by Semenova et al.
(2013) and Lebedeva et al. (2017) based on partial KWBS data set as the following: 140 mm for
larch and swampy sparse growth forest, 110 mm in dwarf cedar and alder shrubs of tundra belt,
and about 70 mm for rocky talus. This database presents full data of observations which allow
for correction of the values as described below.
The highest values of soil evaporation during the summer period were observed at the
larch forest (site 9) and reached 136 mm. At a similar landscape (site 1), this value is lower, at
119 mm, which indicates the influence of local factors. The lowest values of soil evaporation are
104 mm at the plot located at dwarf cedar tree bush (site 7). In July, soil evaporation values
range from 33 to 40 mm, depending on the landscape. In September, the contribution of
evaporation decreases to 14-24 mm (Table 5).
**3.5 Snow evaporation**
Snow evaporation observations were conducted at the KWBS from 1951, but only the
data for the period 1968-1992 is considered to be consistent and reliable and is published in
described database. From 1968 to 1981, the observations were conducted with standard
evaporimeters GGI-500-6 at weather plot Nizhnyaya. In 1981 the snow evaporation observations
were transferred to the Kulu weather plot and lasted till 1992.
Observations of evaporation from snow were made mainly in the fall (September,
October) and in the spring (May - March). During winter months (January - February), the
observations were made only until 1973, because the amount of evaporation from snow proved
to be extremely insignificant for water balance. In the spring, during the intensive snowmelt,
additional weighing of the evaporimeters was carried out every 3-6 hours. In the database, the
evaporation values for night (20-8 hours) and daytime (8-20 hours) intervals are presented, only
those values that correspond to a full 12-hour period of observations are published. The accuracy
is 0.01 mm. The average values of evaporation from snow in mm per day are as follows:
January-February – -0.04; March – +0,09; April – +0,40; May – +0.74; September – +0,20;
October – +0,01. Typical values of evaporation from snow for 1976-1977 are presented at Fig. 8.
**3.6 Thaw/freeze depth**
Since 1952, the observations on permafrost seasonal thaw dynamics were conducted at
the KWBS. Danilin cryopedometers were installed at permafrost observation sites (Snyder et al.,
1971) which mostly were located in the approximate vicinity of Nizhnyaya and Verkhnyaya
weather stations at the slopes with different aspects and landscapes. During 1952-1997 38
cryopedometers were functioning in total. The longest observation period is 33 continuous years
(cryopedometer 17.5 located at the forest with bushes, maximum thawing is 130 cm, 1964-
1997). The deepest values of thawing were observed in rocky talus landscape and can reach
more than 240 cm. The shallowest values of thawing range from 60 to 70 cm at swampy forest.
Thawing of soils at the forest zone varies in large ranges and depends on the location of the
cryopedometer at a slope (Table 6, Fig. 9). Despite the fact that permafrost observation sites
were equipped with special bridges for observers to come close (Fig. 5), eventually surface
damage in the area where the device was installed began to influence thaw depth.
**3.7 Streamflow**
Runoff observations were carried out at 10 catchments: the creek Kontaktovy (the gauges
Verkhny, Sredny, Nizhny), brooks Morozova, Yuzhny, Vstrecha, Vstrecha (the mouth),
Dozhdemerny, Severny, Ugroza (Fig. 10). Key characteristics of the catchments are listed in
Table 7. All the water level gauges were equipped with «Valdai» water level recorders, as well
as needle and hook water level gauges. In spring and autumn, when recorders did not work
properly due to ice on the creeks, discharge was measured more frequently, every 4 hours. To





prevent the recorder floats from freezing, the wells were heated with electric bulbs. At small Morozova and Yuzhny brooks runoff was measured by means of a V-notch weir, at the Severny brook – with a flow measuring flume.

Flow at KWBS begins in May, most of it occurs in summer. For the summer period, rainfall floods are typical (Fig. 11).

In October, small creeks freeze completely. Along the whole length of the Kontaktovy Creek, channel taliks can be found. They go all the way through the layer of alluvial sediments and their depth reaches 15 m in the cross section of the Nizhny hydrological gauge (Mikhaylov, 2013). Taliks freeze in winter only partially.

Annual flow depth of the Kontaktovy stream basin with area 21.3 km$^2$ (average altitude 1070 m) is 281 mm for the period 1948-1997, it increases with the elevation and at the Morozova catchment (mean elevation 1370 m, basin area 0.63 km$^2$) reaches 453 mm (1969-1996). The flow from south-facing (Severny) and north-facing (Yuzhny) micro-watersheds with area of 0.38 and 0.27 km$^2$ are 227 and 193 mm for the period 1960-1997 respectively.

In the database, mean daily values of streamflow are presented.

## 4. Data availability

All data presented in this paper are available from the "PANGAEA. Data Publisher for Earth & Environmental Science" (see Makarieva et al., 2017, https://doi.pangaea.de/10.1594/PANGAEA.881731). The directory includes 12 elements:

1. daily precipitation time series at different gauges within Kolyma Water-Balance Station (KWBS), 1948-1997;
2. daily runoff time series at different gauges within Kolyma Water-Balance Station (KWBS), 1948-1997;
3. evaporation time series at different sites within Kolyma Water-Balance Station (KWBS), 1950-1997;
4. meteorological observations at different sites within Kolyma Water-Balance Station (KWBS), 1948-1997;
5. monthly precipitation time series at different gauges within Kolyma Water-Balance Station (KWBS), 1948-1997;
6. precipitation (10 day sum) time series at different gauges within Kolyma Water-Balance Station (KWBS), 1962-1997;
7. precipitation (5 day sum) time series at different gauges within Kolyma Water-Balance Station (KWBS), 1966-1997;
8. snow survey line characteristics within Kolyma Water-Balance Station (KWBS), 1959-1997;
9. snow survey time series at different sites within Kolyma Water-Balance Station (KWBS), 1950-1997;
10. soil temperature time series at the Nizhnyaya meteorological station within Kolyma Water-Balance Station (KWBS), 1974-1981;
11. thaw depth and snow height time series at different sites within Kolyma Water-Balance Station (KWBS), 1954-1997;
12. evaporation time series from snow within Kolyma Water-Balance Station (KWBS), 1968-1992.

## 5. The future of the KWBS

In summer 2016, with the assistance of Melnikov Permafrost Institute of Siberian Branch of Russian Academy of Science, a group of specialists, consisting of representatives of different scientific institutions, conducted a reconnaissance survey of the KWBS in order to find out if it was possible to carry out scientific research and stationary monitoring of permafrost and



hydrological processes at the station. Despite rather difficult logistic access to the KWBS, it was considered possible to organize accommodation and provision of the station for the period of summer expeditions. At first, the main goal of research resumption at the station would be a renewal of regular observations of runoff, meteorological elements and active layer dynamics at three small catchments (Morozova, Yuzhny, Severny) and the KWBS main-stream outlet (Nizhny gauge) using advanced equipment with automatic data recording. As a result, some unique runoff observations series – over 60 years long – will be continued, which will allow for evaluation of climatic impact on permafrost and provide a scientifically based forecast on current and future climate change impact on the hydrological regime.

During short 3-4 week field trips at the beginning and at the end of the warm (and hydrological) season, it would be also possible to study specific processes of runoff formation under permafrost conditions. Slope runoff occurs unevenly, and is concentrated in particular areas, the drainage zones or preferential path flows. Reconnaissance surveys of the Kontaktovy creek catchment at the KWBS territory, 2016, revealed that there are several types of such zones of slope runoff concentration. Another possible scientific task is to evaluate the role of cryogenic redistribution of runoff, which regularly occurs due to ice freezing-melting in coarse-grained slope deposits. Similar studies have already been carried out in mountain regions of permafrost, including the KWBS (Sushansky, 1999; Bantsekina, 2001; Bantsekina, 2002; Boyarintsev et al., 2006). Another research issue is the study of floodplain taliks (Mikhaylov, 2013) and aufeises (Alexeev, 2016) and their impact on hydrological processes in the mountainous part of the continuous permafrost zone. Field trips for a limited group of scientists could be covered with relatively modest financial support through research grants. In the future, could the aim is for the KWBS to get back its status of a research station, to receive state funding, obtain sponsor support from gold mining companies of the Magadan region and become an international center for complex studies in the field of permafrost hydrology.

The KWBS is situated in the region where monitoring of natural processes is extremely sparse. From 1986 to 1999, the number of hydrological gauge stations in Far-East parts of Siberia decreased by 73% (Shiklomanov et al., 2002). Resumption of water balance observations and organization of complex research of permafrost, climate, and landscape, hydrological and hydrogeological processes based on data collected at the KWBS would make it possible to get new data, representative for the understudied territory of the Arctic in the context of environmental changes. Considering insufficient knowledge about this territory and available long-term data, the KWBS has the prospect to become a highly demanded complex international center for testing natural process models at different scales – from point to regional, – validation of remote sensing products and a place for multidisciplinary field research.

More than 20% of the Northern Hemisphere is covered by permafrost. Three of the four largest rivers of the Arctic Ocean basin flow through Siberia. Many studies highlight ongoing and intensifying changes of water, sediment and chemical fluxes at all spatial scales but mechanisms of changes and future projections are highly uncertain. There are no research centers that could conduct focused studies of hydrological processes at catchments in the permafrost region in Russia. The KWBS incorporation into the international network for monitoring natural processes in cold regions (Interact, SAON, CALM, GTN-P, etc.) could significantly enhance international cooperation for better understanding of cold-region hydrology for the last 70 years, present and future.

Nowadays, the resumption of continuous observations and research at the Kolyma station appears to be a critical task due to increased interest in the natural processes of the Arctic region. Present-day data, following the KWBS long-term observations series, could become a valuable indicator of climate change and a basis for studying its impact on the state of the permafrost and its associated hydrological regime. Currently, as the station infrastructure is still partly intact, and some of the specialists who worked at the KWBS are still active and willing to help, it is necessary to gain attention and support from the Russian and international scientific community regarding the renewal of the KWBS before it is too late.



## 6. Conclusions

The presented dataset describes water balance, hydrometeorological and permafrost related components at small research watershed in mountainous permafrost zone of North-East Russia, the Kolyma water-Balance Station (KWBS). It includes 50 years of continuous daily meteorological and streamflow data for main meteorological plot and runoff gauge of KWBS and daily data of shorter periods for another two meteorological sites and 9 runoff gauges. Meteorological data includes values of air temperature, water vapour pressure, vapour pressure deficit, atmospheric pressure, wind speed, low and total cloud amount, and surface temperature. The dataset also includes all-year daily, warm period daily, 5-, 10-days and monthly sums for 47 (in total) precipitation gauges within KWBS territory for different time spans over the period 1948-1997. It also contains soil evaporation data from different landscapes, snow evaporation series from two sites; snow surveys results for different watersheds within KWBS, as well as thaw/freeze depths at more than 30 observational sites.

The dataset is important because it characterizes the natural settings, which, on the one hand, are nearly ungauged, and on the other hand, are representative for the vast mountainous territory of Eastern Siberia and North-East Russia. It is unique because it combines water balance, hydrological and permafrost data which allow for studying permafrost hydrology interaction processes within the range of all scientific issues, from models development to climate change impacts research.

## 7. Author contribution

O. Makarieva and N. Nesterova digitized and prepared the dataset for publication with assistance from L. Lebedeva and S. Sushansky. The data were collected in 1948-1997 by Hydrometeorological Service of USSR and Russia and published in Observation Reports (1948-1997).

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

Table 1 List of meteorological observation data

| Station | Nizhnyaya | Verkhnyaya | Kulu | Kulu2 |
|---|---|---|---|---|
| Latitude | 61.85 | 61.86 | 61.88 | 61.88 |
| Longitude | 147.67 | 147.61 | 147.43 | 147.44 |
| Elevation, m | 850 | 1220 | 670 | 635 |
| Air temperature | 1948-1997 | 1948-1972 | 1981-1991 1992-1997 | |
| Water vapour pressure | | | | |
| Vapour pressure deficit | 1974-1997 | n/a | | |
| Atmospheric pressure | 1951-1997 | 1951-1972 | | |
| Low cloud amount | 1948-1997 | 1948-1972 | | |
| Total cloud amount | | | | |
| Wind speed | | | | |
| Surface temperature | | | | |

Table 2 Number of snow routes, their total length and number of measurements points –
maximum and minimum values within the whole period of observations

| Watershed | Period | N | L | MP |
|---|---|---|---|---|
| Yuzhny | 1960-1997 | 4 | 1400-1540 | 144-154 |
| Severny | 1950-1997 | 4/10 | 1950/2130 | 23/207 |
| Morozova | 1968-1997 | 2/5 | 960/2645 | 98/534 |
| Ugroza | 1983-1997 | 1 | 1200 | 120 |
| Dozhdemerny | 1959-1971 | 3 | 3240/5720 | 327/575 |
| Vstrecha | 1950-1997 | 1/17 | 2110/10850 | 119/1091 |
| Kontaktovy | 1960-1997 | 2/4 | 4830/13100 | 485/1314 |

N – amount of snow routes; L – total length of the route, m; MP – amount of snow thickness
measurement points.




Table 3 Mean, maximum and minimum observed snow water equivalent (SWE) (mm) before
spring freshet at different landscapes of the Kontaktovy creek watershed, 1960-1997

| Landscape | SWE | | |
|---|---|---|---|
| | mean | max (1985) | min (1964) |
| Forest | 144 | 265 | 79 |
| Dwarf cedar tree bush | 127 | 247 | 39 |
| Rocky talus | 100 | 182 | 46 |
| Boulders | 66 | 127 (1974) | 2 |
| Kontaktovy Creek | 121 | 213 | 59 |



Table 4 Mean, maximum and minimum snow water equivalent (SWE) (mm) before spring
freshet at different watersheds within KWBS

| Watershed | Period | SWE | | |
|---|---|---|---|---|
| | | mean | max | min |
| Yuzhny | 1960-1997 | 121 | 166 | 70 |
| Severny | 1950-1997 | 126 | 232 | 62 |
| Morozova | 1968-1997 | 161 | 298 | 71 |
| Ugroza | 1983-1994 | 133 | 200 | 93 |
| Dozhdemerny | 1959-1971 | 82 | 111 | 53 |
| Vstrecha | 1951-1997 | 123 | 213 | 60 |


Table 5 Mean evapotranspiration (mm) in June – September at different landscapes of KWBS

| # site | Landscape | Period | Elevation, m | Slope aspect | Jun | Jul | Aug | Sep | Total* |
|---|---|---|---|---|---|---|---|---|---|
| 1 | Larch forest | 1962-1997 | 850 | n/a | 35 | 37 | 30 | 17 | 119 |
| 6 | Swampy sparse growth forest | 1969-1982 | 970 | North | 37 | 38 | 30 | 19 | 124 |
| 7 | Dwarf cedar tree bush | 1972-1997 | 1020 | n/a | 30 | 33 | 25 | 17 | 104 |
| 8 | Dwarf cedar tree bush | 1976-1997 | 900 | South | 47 | 40 | 30 | 14 | 131 |
| 9 | Larch forest | 1982-1992 | 669 | West | 36 | 39 | 37 | 24 | 136 |

*the sum for warm period

Table 6 Maximum depth of thawing at the different landscapes

| # site | Watershed | Landscape | Period | Elevation, m | Maximum depth of thawing, cm |
|---|---|---|---|---|---|
| 1 | Kontaktovy | Forest | 1954-1966 | 841 | 150 |
| 6 | Dozdemerny | Rocky talus | 1960-1965 | 1048 | >240 |
| 9 | Severny, Ugroza | Rocky talus | 1954-1966; 1977-1978 | 986 | 168 |
| 12 | Vstrecha, Severny | Dwarf cedar tree bush at rocky talus | 1954-1962; 1966-1968; 1971-1997 | 866 | 157 |
| 15 | Dozdemerny | Dwarf cedar tree bush at rocky talus | 1958-1968; 1970-1982 | 952 | >150 |
| 17 | Vstrecha | Forest | 1960-1965, 1969 | 914 | >124 |
| 18 bh7 | Kontaktovy | Peat bogs | 1959-1960 | 835 | 69 |
| 18 bh8 | Kontaktovy | Peat bogs | 1959-1960 | 835 | 64 |




Table 7 The characteristics of KWBS watersheds

| Code | Catchment (creek – outlet) | Period | Area, km² | X | Y | Stream length, km | Mean watershed width, km | Mean stream slope, ‰ | Mean basin slope, ‰ | Catchment altitude (max-min, mean), m | Mean annual flow, mm | Maximum observed daily discharge, m³s⁻¹ |
|---|---|---|---|---|---|---|---|---|---|---|---|---|
| 1104 | Yuzhny | 1960-1997 | 0.27 | 61.84 | 147.66 | 0.51 | 0.35 | 235 | 303 | 1110-917, 985 | 193 | 0.14 |
| 1107 | Severny | 1958-1997 | 0.38 | 61.85 | 147.66 | 0.74 | 0.38 | 175 | 388 | 1300-880, 1020 | 227 | 0.18 |
| 1103 | Morozova | 1968-1996 | 0.63 | 61.84 | 147.75 | 0.97 | 0.45 | 326 | 649 | 1700-1100, 1370 | 453 | 0.44 |
| 1624 | Ugroza | 1983-1991 | 0.67 | 61.86 | 147.67 | 0.9 | 0.74 | 218 | 461 | 1270-914, 1260 | 354 | 0.27 |
| 1106 | Dozhdemerny | 1952-1971 | 1.43 | 61.86 | 147.63 | 0.87 | 0.99 | 220 | 432 | 1450-950, 1180 | 208 | 0.31 |
| 1105 | Vstrecha | 1949-1997 | 5.35 | 61.85 | 147.66 | 3.4 | 1.5 | 92 | 346 | 1450-833, 1060 | 237 | 3.15 |
| 1100 | Kontaktovy – Verkhny | 1973-1980 | 5.53 | 61.84 | 147.70 | 2.8 | 2.1 | 185 | 473 | 1700-909, 1070 | 317 | 2.52 |
| 1625 | Vstrecha – the mouth of Ugroza Cr. | 1984-1996 | 6.57 | 61.84 | 147.66 | 3.6 | 1.8 | 76 | 406 | 1450-831, 1070 | 283 | 2.6 |
| 1101 | Kontaktovy – Sredny | 1948-1997 | 14.2 | 61.84 | 147.67 | 6.2 | 2.8 | 65.2 | 413 | 1700-842, 1120 | 289 | 7.02 |
| 1102 | Kontaktovy – Nizhny | 1948-1997 | 21.3 | 61.85 | 147.65 | 7.1 | 3.7 | 57.6 | 413 | 1700-823, 1070 | 281 | 8.15 |


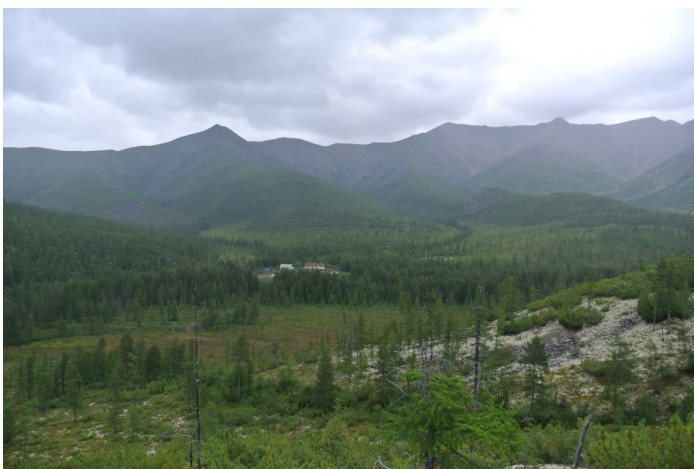


Fig. 1 The view of the Kolyma Water Balance Station, August 2016 (photo by O. Makarieva)



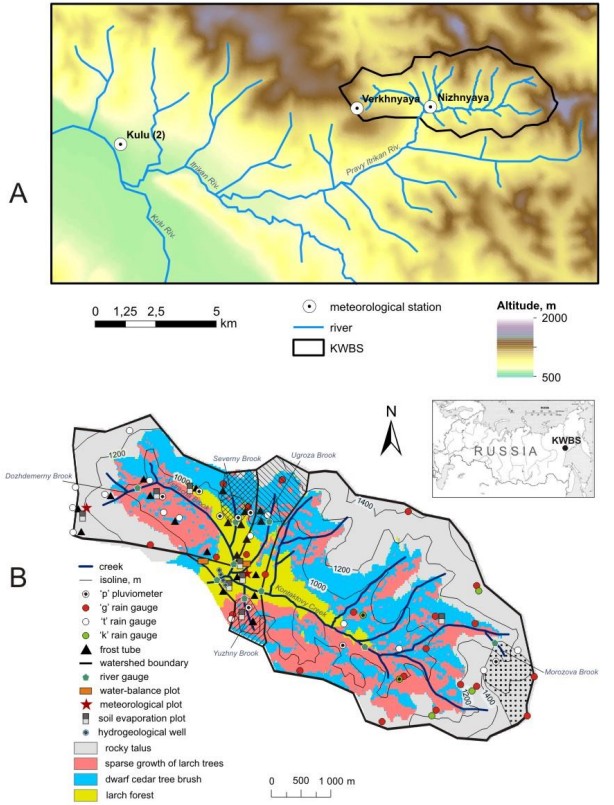


Fig. 2 Scheme of the Kolyma Water Balance Station (KWBS) indicating the location of observation sites


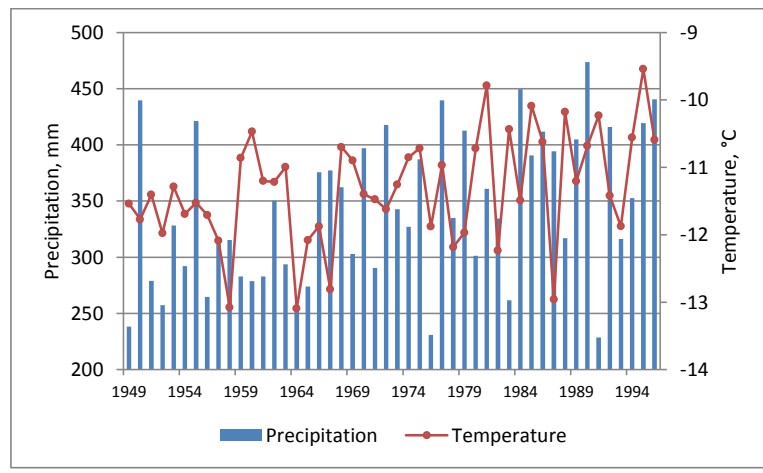


Fig. 3 Annual precipitation (mm) and air temperature (°C) at Nizhnyaya weather station, 1949-1996






Fig. 4 Mean monthly precipitation (mm) and air temperature (°C) at Nizhnyaya weather station, 1949-
1996

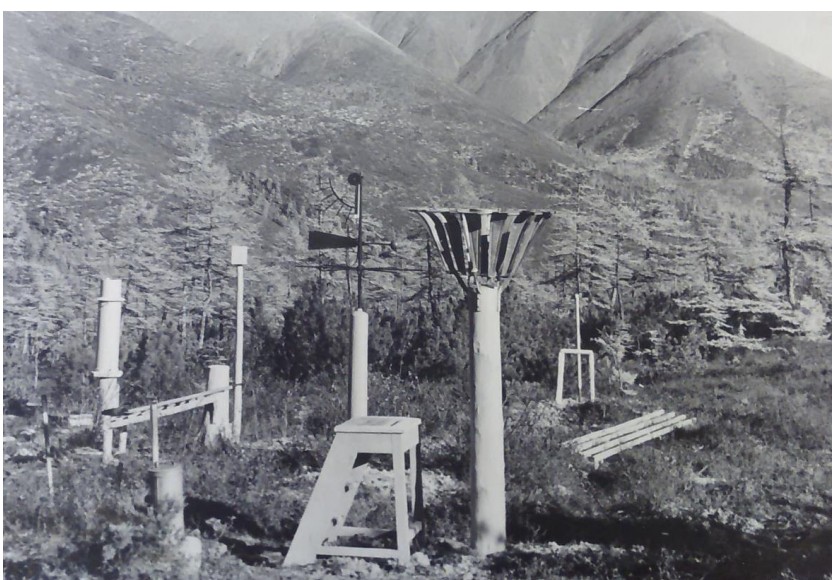

Fig. 5 Meteorological observations: rain-gauge plot #2, 1959.

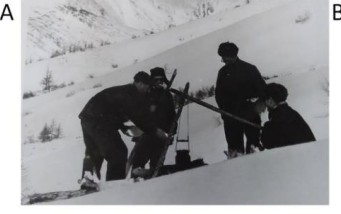 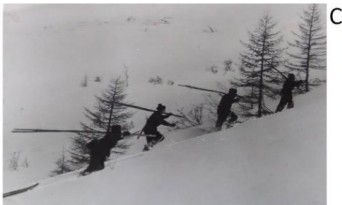 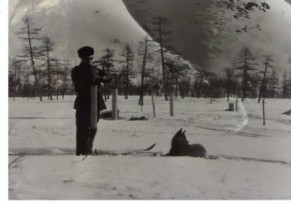

Fig. 6 Snow survey at the Kontaktovy creek catchment (A, B) and measurement of snow density (C),
1960






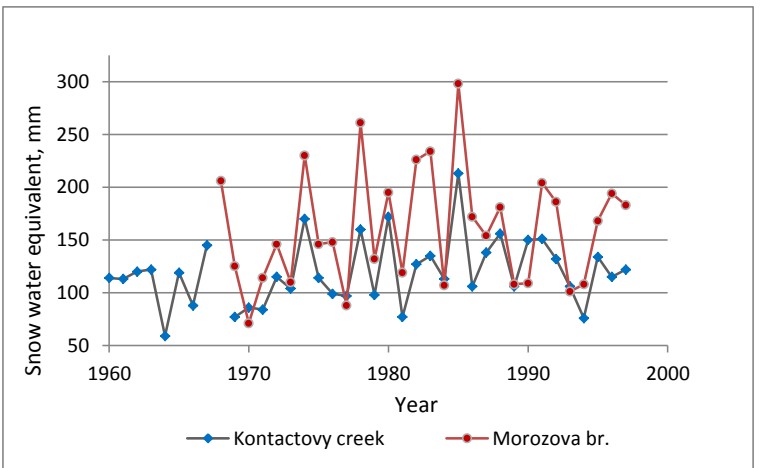

Fig. 7 Snow water equivalent at the Kontaktovy creek and Morozova br.


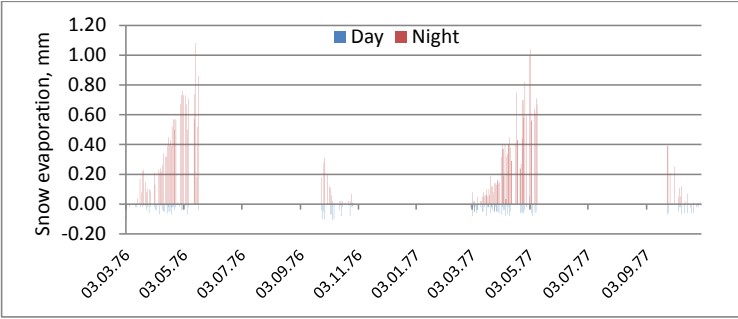


Fig. 8 Snow evaporation (mm) during day and night period, 1976-1977





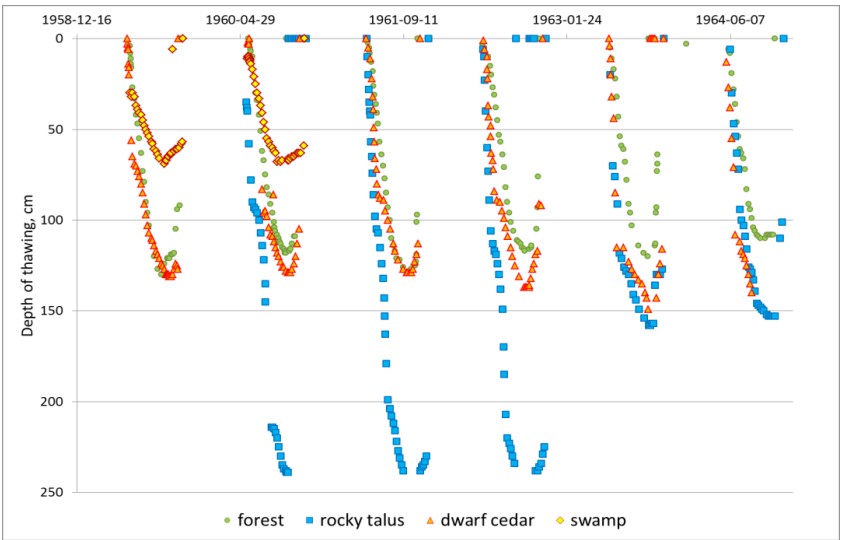

Fig. 6 Depth of ground thawing at the different landscapes of KWBS


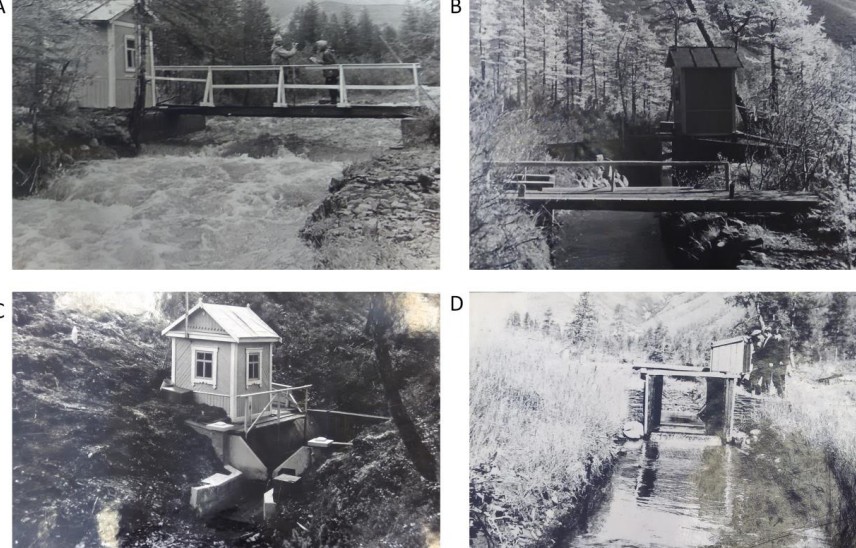

Fig. 7 Runoff observations: A – runoff gauge at the Kontaktovy creek, 1979; B – runoff gauge at the
Dozhdemerny creek, 1959; C – runoff gauge at the Yuzhniy creek, 1960; D – runoff gauge at the
Vstrecha creek, 1953.





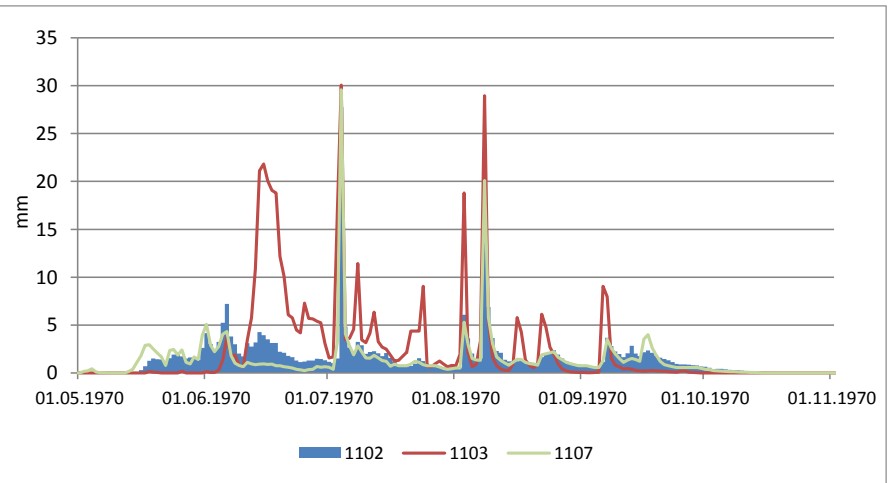


Fig. 11 Flow depth (mm) at the Kontaktovy creek – Nizhny (1102), Severny br. (1107) – south-facing
slope with cedar dwarf bush landscape and Morozova br. (1103) – rocky talus landscape at watershed
divides, 1970