# Peer review of "Water-balance and hydrology research in a mountainous permafrost watershed in upland streams of the Kolyma River, Russia: a database from the Kolyma Water-Balance Station, 1948-1997"

_Earth System Science Data, 2017_

## Short Comment (SC1) · 21 Nov 2017

The importance of the database can not be overemphasized! Thanks to the authors!

---

## Short Comment (SC2) · 3 Dec 2017

The paper gives a concise overview of the data that where collected at the KWBS during the period 1948-1997 (partly later too). It also gives some insight into the difficult and harsh conditions that the observers and scientists and were facing. Considering that conditions and the comprehensive instrumentation, the data collection is a remarkable piece of work, which deserves both attention and willingness to follow up. In this

way a first important step has been made; the digitizing, organizing and publication of the data. The possibility for free access will give other scientists the opportunity to conduct new studies, test their models and so on. The impressive density of precipitation gauges within the catchment for instance, would be a very good basis to study precipitation distribution and gradients. The historical data, together with the resumption of the data collection at the KWBS would after some years give an unique possibility to compare climate model results with real measurements in a high arctic permafrost region. In general, not many data series with such a long duration and for such a wide variation of hydrometeorological variables are available. As a hydrologist, I really would like to see that the authorities, or other interest groups, would acknowledge the importance of such hydrological data, especially considering how spares data in that region of the earth are. Of course there will be the need to invest some money, but I think that money would give valuable payback in form of gained knowledge, better understanding and hopefully better preparedness for the ongoing and coming changes in the climate. The estimation of the value of hydrological data has always been, and will further be an issue. Of course approaches have been made (there is a number of publications an that topic), but I am not sure if the message is reaching the right people. In this sense it is up to us hydrologists to spread the message, and this is what the authors of the paper are doing in their approach to initiate the resumption of the KWBS. I have not been looking closely at the data, but at glance, I see that there is some short data description, following the data. Further use of the data might lead to the need for a more detailed description of the data quality. That could be implemented in the database. I also have some small comments and suggestions considering the text in the paper. I will send these comments directly to the corresponding author. I wish the authors and other involved parties good luck with their further work and hope they will succeed.

---

## Referee Comment (RC1) · Anonymous Referee #1 · 5 Dec 2017

This manuscript by Makarieva and co-authors presents an extraordinary hydrological dataset (including its history) from a permafrost region in the North-East of Russia. It includes meteorological variables, as well as records of runoff, snow and soil frost from different locations within watersheds of 0.27 to 21 km2 area collected between 1947 and 1997. The objective of this manuscript is to make the research community aware of this valuable data treasure and to give access to the data. I agree with the authors that the presented data set, representing a part of the world with very limited hydrological longterm information, is unique and therefore of great importance in the context of climate change. Therefore, I highly appreciate this publication and hope to see it published in Earth Syst. Sci. Data, which seems to be an appropriate journal for such a data-announcement paper. The provided information is clear and well-structured for most of the part. However I have one major and a few minor questions and suggestions that may help improving the manuscript.

PS: I've made random checks of the dataset. It was easily accessible and well-structured and documented. The quality of the data seemed to be good.

Major comment:

The discussed data set is called "Kolyma water-balance station" implying that the water balance has been calculated here. However, only the different components of the water balance are presented in this manuscript, but not the balance. I know, it is very difficult to do that for such a location where winter conditions are extremely harsh. Nevertheless, I wonder whether or not the annual water balance has been calculated and suggest to include such results if available. Else you may explain why the term "water balance" is so prominent in the name of the station.

Minor comments:

- In the abstract you state that "the data are representative for the vast territory of the North-East of Russia". What is this claim based on?

- You mention different types of evaporimeters (Rykachev, Gorshenin, GGI-500), some of them used also for snow evaporation measurements. Could you briefly explain how these evaporimeters differ from each other. Also, when you measure snow evaporation, is this representative for snow on the ground? I assume (and know from our own longterm observations) that snow evaporation from trees (interception) is more relevant than from the ground. Can you comment on that?

- I guess that most readers of this journal don't know what a Danilin cryopedometer is.
I understand that it measures the thaw depth of the active layer, but how exactly?

- Some of the figures are just too small. For example, the map of Fig.2 should be enlarged to become readable. Also the very nice photos in Fig. 6 showing the hard work of measuring snow would deserve a larger format.

- Different types of rain gauges and shields are mentioned in section 3.2 (Nipher, Tretyakov, Kosarev and pit rain gauge), and a reference is made to Fig. 5. But which one of these types are actually shown in Fig. 5? Please clarify in the figure caption. And, if possible, could you shortly explain how these specific types differ from each other.

- line 260: "correspond to" instead of "account for"

- line 275: "amount to" instead of "account for"

- line 309: add "the" after "published in"

- line 415: remove "could"

- Table 2: you may include the lower caption into the table

---

## Short Comment (SC3) · 13 Dec 2017

This paper presents an excellent description of the Kolyma Water Balance Station and the observations that were collected at the station from 1947 to 1997. Given the location of the station and the sensitivity of continuous permafrost to climate warming, it would be a loss for science if this station is not restored to full operation. The authors make a good suggestion about the possibility of the station being available during

part of the year for in-residence research from international scientists. But, even if that recommendation is not implemented due to the costs necessary to have suitable accommodations, I strongly recommend that funding for the station be re-stablished to allow the continuation of this priceless dataset.

Editorial comments/suggestions:

51: it would be good to have the geographic coordinates at the beginning.

163 and 164: Do you mean "…Osipiev AND the hydrologist-technician…"?

166: what does it mean partially? Only part-time?

170: what do you mean by rationalization of the network?

172: what are the sensors included in the water balance sites?

196: max and min temperature?

214: cliff? Do you mean slope? Cliff is almost vertical, too steep to have a station there.

223: What do you mean by "The angle of the horizon is 4 224 degrees"?

415: eliminate "could" from the sentence

425: "Considering the insufficient…"

Figures 5, 6 and 9 should have a source cited. Also, Figure 9 appears as Figure 7 in its caption.

---

## Referee Comment (RC2) · Anonymous Referee #2 · 19 Dec 2017

Water-balance and hydrology research in a mountainous permafrost watershed in up-land streams of the Kolyma River, Russia: a database from the Kolyma Water-Balance Station, 1948-1997

General comments: The manuscript presents a unique data set from a permafrost region on meteorological parameters, soil frost data and hydrological parameters. There is a general lack of well documented and comprehensive hydro-meteorological

datasets (with sufficiently long time series) from permafrost regions and this fact solely motivates the publication of the present study. In times of climate change and discussions of impact on hydrology due to warming permafrost the base line data is of extra importance. Thus, the fact that only historical data is presented here is not a problem, it can be used in the scientific community in order to better understand the permafrost-hydrological system pre warming. The manuscript is well written and structured, however the authors should consider doing some structural edits according to suggestions below. Also, the title of the manuscript is misleading the reader since no water balance is presented for the study site. The different components of the water balance is presented, but no suggestions on how to set up the WB is given. I would recommend to change the title in order to better describe what is included in the manuscript.

Specific comments: 1. Introduction: I recommend to go through the already published data sets in ESSD related to hydrological data in permafrost and arctic areas. It would be nice to get a more thorough picture of available data and how the data in the present manuscript complement already published hydrometeorological data from the arctic regions.

2. Site description: The permafrost conditions is described. How about taliks in the area? Taliks have great impact on the interaction between permafrost and hydrological flows, describe shortly the presence of taliks in the areas and where they are found (under lakes or rivers) and what type of talik that is most common (open, close, through)

3. Data description: The data description and main results are given in the same section. I would recommend the authors to separate the technical description of equipment, installation techniques, measured time periods etc from result presentation of the collected data. A new chapter 4 presenting the main results for each parameter should facilitate for the reader. Inter- and intra annual patterns in the data should be presented in the results section and not in section 3 as it is presented in the present version of the manuscript.

[Figure]

4. A table early in chapter 3 summarizing the measured parameters including details of measurement period, periods of data gaps and used equipment and/or methods for evaluation of data would give a better overview of the presented data, reference to observation points in the map in Figure 2 could also be listed in the table.

5. There is no or very little information about uncertainties and accuracy for the equipment used in the investigations. If information is available (given that the measurements were performed long time ago and technical descriptions of used equipment can be hard to find) a complementary section about uncertainties would rise the quality of the manuscript.

6. Precipitation data: No details are given about the correction of precipitation data. I guess the data presented are uncorrected for wind and adhesion losses. Given that much of the precipitation fall as snow, the under-catch might be high and the errors due to this have to be discussed. Motivate why data is not corrected and provide the reader with necessary information about the location of the precipitation bucket/meteorological station to a proper correction can be made. The under-catch in wind exposed areas can be as high as 30-40% during the snowy seasonRefernces to methods for correction and how this has been handled in other hydrometoerological studies should be given.

7. The data in Pangaea: A complemtary data set with maps in ArcGIS format would facilitate the use of the data-set in future studies. A base set of catchment geometries, land use, soil distribution, location of lakes and rivers, topography etc would make it much easier for data-users to set up proper hydrological models of the study site.

8. Tables: The sites referred to in the tables are in general hard to find in the map in Figure 2. A clear coupling between site ID and the map must be given. The map, including the labels and legend, have to be enlarged. 9. Row 261: "Snow cover at KWBS is formed in the first weeks of October"...based on data for which period? Give correct reference. 10. Row 271: How is the SWE quantified? By weighing the snow or by calculation?

---

## Author Comment (AC1) · 10 Feb 2018

Dear Michael! Thank you for your encouragement and help when we are in Magadan. Currently when the network of hydrometeorological observations has diminished catastrophically in the North-East of Russia, the data of the Kolyma Water Balance station is priceless. Considering the fact that meteorological station and one runoff gauge are still in use, the length of continous time series is almost 70 years which is quite unique

for mountainous permafrost research watershed. Next step shpuld be the analysis and publication of the data of last 20 years.

---

## Author Comment (AC2) · 10 Feb 2018

We are grateful to Wolf Marchand for understanding the importance of resumption the observations at the Kolyma Water-Balance station and his support in this issue. We are trying our best to convince the authorities of the Magadan district of Russia to start the research program at the station while the station infrastructure is still partly intact, and some of the specialists who worked at the KWBS are still active and willing to help.

[Figure]

The small changes to the text, mostly regarding English and erratums, were made accordingly to the comments of Dr Marchand which he sent to the main author of the manuscript personally.

---

## Author Comment (AC3) · 10 Feb 2018

We appreciate the comments of Dr Restrepo and his support of the idea of the Kolyma station reviving. Below we provide the corrections and responces which were made according to Dr Restrepo's comments.

Editorial comments/suggestions: 51: it would be good to have the geographic coordinates at the beginning. The coordinates were added (line 49)

and 164: Do you mean ". . .Osipiev AND the hydrologist-technician. . ."? Yes, exactly. Corrected accordingly (line 195)

166: what does it mean partially? Only part-time? Clarified, line 197

170: what do you mean by rationalization of the network? We changed from rationalization to optimization meaning the establishment of rain gauges in new locations and shutting down some non-representative rain gauges (lines 201-203)

172: what are the sensors included in the water balance sites? Water-balance plot is the part of the slope bordered by walls and equipped with devices to intercept and account for surface and subsurface runoff and soil erosion. The plots were equipped with the trays which allowed for flow accounting from surface and different soil horizons. Water from the water intake trays enters the measuring pavilion, where flow is measured in tanks equipped with ÂńValdaiÂż water level recorders, as well as needle and hook water level gauges. Additionally precipitation was measured by Tretyakov gauge and pluviometer, also snow depth and density at several points, and thaw/freeze soil depth. In presented database we do not publish the data of water-balance plots, therefore their description is skipped.

196: max and min temperature? No, only mean daily air temperature was published in the Observation Reports and is presented in the database.

214: cliff? Do you mean slope? Cliff is almost vertical, too steep to have a station there. Cliff is changed to slope (line 246)

223: What do you mean by "The angle of the horizon is 4 224 degrees"? This is the special characteristic of the location of meteorological station describing its position towards large obstacles (buildings, forest, hills). It is used in Russian climatology for correction or estimation of such meteorological elements as wind, sun radiance, etc. We eliminated this characteristic from the text because it is too specific and would require long explanation.

415: eliminate "could" from the sentence Corrected

425: "Considering the insufficient. . ." Corrected

Figures 5, 6 and 9 should have a source cited. All captions of the figures containing historical photos are corrected by citing the source of photos.

Also, Figure 9 appears as Figure 7 in its caption. Corrected

You use "till" instead of "until". While "till" is correct, it is every informal. I recommend that you change it to "until" on technical papers. It is similar to using "gonna" instead of "going to" Corrected

Please also note the supplement to this comment:
https://www.earth-syst-sci-data-discuss.net/essd-2017-125/essd-2017-125-AC3-supplement.pdf

[Figure]

**Supplement:**

[revised manuscript text omitted]

Though in the last several decades and more recently many research watersheds were established in Arctic zone of the USA and Canada, to the best of our knowledge, the first systematic cold-region hydrology observations in North America began not earlier than the 1960s. Such, the Caribou-Poker Creeks Research Watershed was established only in 1969 (Hinzman et al., 2002), 20 years later than KWBS.

One may mention numerous scientific catchments in Alaska – Fish Creek (Pacific Northwest…, 2014), Toolik station (Hoobbie et al., 2003), Tanana River (Yarie et al., 1998), Kuparuk River (Arp & Stuefer, 2017, Kane & Hinzman, 2009), Imnavait River (Walker & Walker, 1996), Putuligayuk River (Kane & Hinzman, 2009), as well as Arctic monitoring programs (NPR-Hydrology (NPR-A Hydrology…, 2018), Arctic Observatory Network (Arctic Observatory Network…, 2018).

The studies at research watersheds of Canada are integrated into scientific programs and accompanied by data analysis and models development and applications For example, the Changing Cold Regions Network project (Changing Cold…, 2018) includes field studies on 14 watersheds and the use of two Canadian models CHRM and CLASS. The Improving Processes & Parameterization for Prediction in Cold Regions Hydrology (IP3) project (Improving

Processes…, 2018) combined 10 research watersheds and four hydrological models – CHRM (Pomeroy et al., 2007), CLASS (Verseghy, 1991), MESH (Pietroniro et al., 2007), GEM (Yeh et al., 2002).

Although there are large mountainous areas in other cold regions of the world, the combination of extremely severe climate (mean annual air temperature reaches -11.3°C) and continuous permafrost creates unique conditions at KWBS which are not presented at any other research watershed of the world.

Nasybulin (1976) showed that hydrological regime at the KWBS is representative for the whole Upper Kolyma Plateau. Taking into account the similarity of main landscape types across mountainous regions of North-East Russia to those found at the KWBS, the conclusion can be made that hydrological conditions at KWBS are actually representative for vaster ungauged areas, than described by Nasybulin (1976).

Sufficiently long time series of observations which were continuously conducted by uniform methods and covered pre-warming period are of high importance for the studies of climate change impact on hydrology in the Arctic.

[revised manuscript text omitted]

The list of used devices and the accuracy of observations for each meteorological element is presented in the description files of the database.

**3.2 Precipitation**

In total, the precipitation was observed at 47 gauges within KWBS territory during different periods. Continuous daily all-year around precipitation data is available for the period 1948-1997 for the gauge (#12) at meteorological station Nizhnyaya and for the gauge (#54) at meteorological station Kulu for 1981-1997. Four gauges have the data of daily totals during warm season for the period for more than 30 years and another 18 gauges for different shorter periods. Usually the start of daily observations at those gauges was initiated by the beginning of snowmelt period and lasted until the end of September. Monthly sums of precipitation were measured at 30 gauges, 10-days and 5-days sums – at 21 and 18 gauges respectively.

In 1948 precipitation gauge stations for measuring daily precipitation were equipped with the Nipher-shielded and Tretyakov-shielded precipitation gauges (Fig. 3A). In 1948-1958 the observations were carried out with both devices in parallel, after 1959 only Tretyakov gauges were used. Tretyakov precipitation gauges were also used for measurements precipitation totals in 5 and 10 days periods.

The other types of precipitation gauges applied at the KWBS are the Kosarev and ground rain gauge (GR-28) (Fig. 3B, 3C). GR-28 gauge with receiving area 500 cm$^2$ was installed into the special box several cm above the ground. GR-28 were usually installed on the 1$^{st}$ of June and dismantled on the 1$^{st}$ of September and used for rain measurements over the longer period, typically one month. Only those GR-28 which were installed at the soil evaporation plots measured precipitation every day. The Kosarev precipitation gauges were used for monthly precipitation measurements from the 1$^{st}$ of October to the 1$^{st}$ of May. Different precipitation gauges are shown at Fig. 5.

In 1960-1963 there was an attempt to register precipitation with automatic radio-precipitation gauges, but due to improper performance of the devices those observations were stopped.

In 1988, precipitation observations at the KWBS were carried out with 36 precipitation and rain gauges, distributed relatively evenly throughout the area and altitudinal zones. Average density of the precipitation network at that time accounted for 1.6 units per 1 km$^2$.

For the period 1948-1968 precipitation data was published in Observation Reports as it was without any correction. Starting from 1969, all daily, 5-days and 10-days totals data from

Tretyakov rain gauges have been corrected for wetting losses according to Manual (1969). The correction value for precipitation event varied from 0.0 to 0.2 mm depending on the amount of observed precipitation and weather conditions. In average annual value of wetting losses correction did not exceed 5 % of total amount of precipitation, though in some years it could reach up to 9-10 %. In 1948-1983 monthly sums data obtained from GR-28 and Kosarev gauges were published without any correction. In 1984 wetting losses correction was introduced to GR-28 observations as well. In the database the precipitation data is presented in original form without any changes.

The analysis of water balance, climate change impact on river runoff or hydrological modelling requires accurate and reliable precipitation data. Arctic and mountainous regions are characterized by high bias of precipitation measurements because of significant amount of snowfall precipitation (WMO Report #67, 1998). Monthly estimates of this bias often vary from 5% to 40%. Biases are larger in winter than in summer largely due to the deleterious effect of the wind on snowfall (Groisman et al., 2015). Three main methods of winter precipitation bias correction are suggested for the Tretyakov gauge which was the main type of precipitation gauge at KWBS. They are the WMO methodology (Yang and Goodison et al., 1995), Northern European countries method (Forland et al., 2000) and the approach developed by Golubev (WMO Report #67, 1998). The basis of all three methods for correcting measured precipitation is the dependence of the aerodynamic coefficient on wind speed, air temperature, precipitation type and wind protection.

In described database each precipitation gauge (if it was available) has the description of its location, altitude, slope exposure, vegetation type. Additional characteristic is the degree of protection characterized by five types of Schwer (1976) classification (Ia, Ib – protected; IIa, IIb – half-protected; III – open; IV – shore station). The database also contains the series of daily wind speed for three meteorological stations which combined with the information on location gauges can be used as a proxy for introducing bias corrections.

**3.3 Snow surveys**

Snow cover observations were started in 1950 and initially conducted at two catchments, at two meteorological plots and four typical squares. In 1959-1960 the number of catchments with snow surveys reached five. Up to 1971, snow surveys were conducted once per month starting in November and finishing in May at small catchments (the Severny, Yuzhny, Dogdemerny and Vstrecha) and once before spring snowmelt at the Kontaktovy – Nizhny. Since 1972, the observations were reduced to one survey per year (usually at the end of April) for all watersheds. Table 2 shows the number of snow routes, their total length and number of measurement points, including their distribution among different landscapes of the catchments (Fig. 4). Snow depth was measured every 10 m, snow density – every 100 m at most of the watersheds, and 5 and 50 m respectively at the Morozova brook watershed.

Based on the data about measured snow height and snow weight with the account for landscape and elevation distribution average SWE for individual watersheds and landscapes was calculated and published in the Observation Reports.

Average depth of snow cover is presented with accuracy of 1 cm, density and SWE – 0.01 $g\,cm^{-3}$ and 1 mm respectively.

**3.4 Soil evaporation**

Three types of evaporimeters were used at the KWBS.

Evaporimeter GGI-500-50 (later modified to GGI-500-30) is a standard device for the soil evaporation measurements in Russia and former USSR (Fig. 5B). It consists of two cylindrical vessels, one inside the other, and a water-collecting vessel. The bottom of the inner cylinder has openings; the core sample is placed in it. The quantity of water evaporated is determined from the difference in weight of the sample as measured over two successive observation periods.

Rykachev evaporimeter was used for the soil evaporation measurements in 1950s. It consists of a sealed square rectangular box with a core sample (Fig. 5A). The box was placed inside another box installed in the ground. Since the inner box was sealed the device did not allow for water infiltration (Chebotarev, 1939).

The description of Gorshenin evaporimeter was not found.

From 1950 to 1953, five soil evaporation plots were opened at the sites with diverse underlying surfaces and different expositions and altitudes. The plots were equipped with the Rykachev and Gorshenin evaporimeters with evaporation area 1000 cm$^2$. The observations until 1958 are considered to be approximate due to the absence of accompanying rain gauges and scales of required accuracy.

From 1958 to 1966, the measurements were conducted at the soil evaporation plot, located near the Nizhnyaya weather station. The observations of evaporost were carried out with two evaporimeters GGI-500 and the Rykachev evaporimeter, precipitation − with a ground rain gauge.

Three soil evaporation sites were established in different landscapes − in the Vstrecha (1967), Morozova (1971) and Yuzhny (1977) brooks basins. The measurements were carried out with standard weighing evaporimeters GGI-500-50, which, due to the physical proximity of permafrost, were changed to GGI-500-30, meaning that their height was decreased to 30 cm.

The accuracy of observations was 0.1 mm (Konstantinov, 1968).

**3.5 Snow evaporation**

Snow evaporation observations were conducted at the KWBS from 1951, but only the data for the period 1968-1992 is considered to be consistent and reliable and is published in the described database. From 1968 to 1981, the observations were conducted with standard evaporimeter GGI-500-6 at weather plot Nizhnyaya. In 1981 the snow evaporation observations were transferred to the Kulu weather plot and lasted until 1992.

This measurement accounts for the snow evaporation on the ground. In the conditions of the KWBS with the larch as the main tree type, intercepted snow was only temporary phenomena because of cyclonic activity in January and February. Wind during the cyclones blow away snow from all trees except dwarf cedar that is under snow for the most part of the winter.

Observations of evaporation from snow were made mainly in the fall (September, October) and in the spring (May − March). During winter months (January − February), the observations were made only until 1973, because the amount of evaporation from snow proved to be extremely insignificant for water balance. In the spring, during the intensive snowmelt, additional weighing of the evaporimeters was carried out every 3-6 hours. In the database, the evaporation values for night (20-8 hours) and daytime (8-20 hours) intervals are presented, only those values that correspond to a full 12-hour period of observations are published.

The accuracy of measurements is 0.01 mm.

**3.6 Thaw/freeze depth**

Since 1952, the observations on permafrost seasonal thaw dynamics were conducted at the KWBS. Danilin cryopedometers (frost tubes) were installed at permafrost observation sites (Snyder et al., 1971) which mostly were located in the approximate vicinity of Nizhnyaya and Verkhnyaya weather stations at the slopes with different aspects and landscapes.

Cryopedometer designed by A.I. Danilin consists of a rubber tube 1 cm in external diameter and calibrated to an accuracy of 1 cm. The tube is filled with distilled water, closed at both ends and lowered into a casing (an ebonite pipe) installed in a borehole in the soil. In order to measure the depth of freezing, the rubber tube is taken from the casing and the lower end of the ice column in the tube is determined (Manual, 1973).

Despite the fact that permafrost observation sites were equipped with special bridges for
observers to come close, eventually surface damage in the area where the device was installed
began to influence thaw depth (Sushansky, 1988).
During 1952-1997 38 cryopedometers were functioning in total.

**3.7 Streamflow**
Runoff observations were carried out at 10 catchments: the creek Kontaktovy (the gauges
Verkhny, Sredny, Nizhny), brooks Morozova, Yuzhny, Vstrecha, Vstrecha (the mouth),
Dozhdemerny, Severny, Ugroza (Fig. 6). Key characteristics of the catchments are listed in
Table 7.
All the water level gauges were equipped with «Valdai» water level recorders, as well as
needle and hook water level gauges. In spring and autumn, when recorders did not work properly
due to ice on the creeks, discharge was measured more frequently, every 4 hours. To prevent the
recorder floats from freezing, the wells were heated with electric bulbs.
At the micro-watersheds of Morozova and Yuzhny brooks runoff was measured by
means of a V-notch weir, at Severny brook – with a flow measuring flume.
In the database, mean daily values of streamflow are presented.
Originally daily discharges were published in Observation Reports in $l\,s^{-1}$ with accuracy
of three significant figures, but not more accurately than $0.01\,l\,s^{-1}$ for runoff gauges equipped
with weir or flume. For gauges with a natural channel for discharges more than $1000\,l\,s^{-1}$ the
rounding to three significant figures was performed, for discharges less than $100\,l\,s^{-1}$ – to two
significant figures, but not more precise than $1\,l\,s^{-1}$.
Small discharges which are less than $0.05\,l\,s^{-1}$ for the gauges equipped with hydrometric
facilities and less than $0.5\,l\,s^{-1}$ for larger watersheds gauges were published in Observation
Reports as 0.00 and 0, respectively. The periods with no runoff because of drying and freezing
were marked with special symbols.
In the database, water discharges are converted to $m^3\,s^{-1}$, the number of significant figures
was preserved but the values 0.00 and 0, as well as special symbols for freeze and dry periods
are indicated as 0.
In 1984-1997 the information on the accuracy of discharge data was published for several
runoff gauges In Observation Reports. It included the percentage of stage curve extrapolation in
both directions which was published for one or several runoff periods per year depending on how
many stage curves were applied. Also information about measured and estimated instant
maximum and minimum discharges was available for the same period. Fig. 7 shows the boxplots
of these characteristics for 7 runoff gauges for the period 1984-1997.

**4. Results**
**4.1 Meteorological variables and precipitation**
The climate of the study area is severely continental with harsh long winters and short but
warm summers. Average annual temperature at the Nizhnyaya meteorological plot during 1949-
1996 is -11.3 °C. Mean monthly temperature in January was -33.6 °C, in July +13.2 °C (Fig. 8-
9). The absolute minimum daily temperature of -53.0 °C was registered in 1982 and the absolute
maximum daily temperature was +22.8 °C (1988). The period of negative air temperatures lasts
from October to April, freeze-free period is, on average, 130 days long.
Air temperature inversions are observed at the KWBS. In December air temperature
gradient reaches +2.0, in May it accounts for -0.5°C per 100 m of elevation respectively.
The average air humidity at the Nizhnyaya station is 3.6 mb, reaching its maximum and
minimum values of 9.8 and 0.4 mb in July and December respectively.
Total cloudiness at the Nizhnyaya station has average annual value of 7.0 and does not
change considerably through the year. Its minimum and maximum values are 5.9 and 8.0 points
in March and July. Lower cloudiness dynamic is more significant, its mean monthly values
changes from 0.5 to 4.7 in March to July with average value of 2.2 points.

Mean wind velocity is more than twice higher at Verkhnyaya station (1220 m a.s.l.) in
comparison with Nizhnyaya station (850 m a.s.l.) and amount to 3.0 and 1.3 m s$^{-1}$ accordingly.
Average monthly values changes from 0.83 in December to 1.70 in May at Nizhnaya, and from
2.7 in November, February to 3.4 in May. Maximum daily wind speed amounted to 16 and 36 m
s$^{-1}$ at Nizhnyaya and Verkhnyaya.

Precipitation at Nizhnyaya meteorological plot from 1969 (the year when wetting losses
were introduced) to 1997 varied from 229 (1991) to 474 (1990) mm per year with mean value of
362 mm. After introducing wetting losses correction to the period 1949-1968 and computing
average mean amount from 1949 to 1997, its value decreased to 351 mm. Maximum and
minimum monthly amount of precipitation at Nizhnyaya station was observed in July and March
and correspond to 72 and 8 mm respectively for the whole period of observations 1949-1997
(Fig. 8-9).

Maximum daily amount of precipitation at Nizhnyaya station was observed in June 1968
reaching 48.1 mm. In average for the period of 50 years this statistic amounted to 26 mm.

**4.2 Snow cover**

Stable snow cover at KWBS in average is formed in the first weeks of October, and melts
in the third week of May (1949-1996). The KWBS area is characterized by an increase in the
thickness of the snow cover due to the absence of thaws during the whole snow season. In the
open treeless and watershed divide areas, the redistribution of snow pack due to wind blow is
observed.

Average for the watershed mean, maximum and minimum snow water equivalent (SWE)
before spring freshet estimated based on snow survey results at the Kontaktovy – Nizhny amount
to 121, 213 (1985) and 59 (1964) mm respectively in the 1960-1997 period. In general, rocky
talus and tundra bush landscape are characterized by lower SWE due to wind blowing. Much
snow is accumulated in the forest landscape. However, at the Morozova brook watershed which
is fully covered by rocky talus landscape, mean SWE before snowmelt was estimated as 161 mm
with the maximum value of 298 mm observed in 1985 reaching in average 0.99 m snow height
(Table 3, Table 4, Fig. 10).

**4.3 Soil and snow evaporation**

The highest values of soil evaporation during the summer period were observed at the
larch forest (site 9) and reached 136 mm. At a similar landscape (site 1), this value is lower, at
119 mm, which indicates the influence of local factors. The lowest values of soil evaporation are
104 mm at the plot located at dwarf cedar tree bush (site 7). In July, soil evaporation values
range from 33 to 40 mm, depending on the landscape. In September, the contribution of
evaporation decreases to 14-24 mm (Table 5).

Average values of annual soil evaporation were previously estimated by Semenova et al.
(2013) and Lebedeva et al. (2017) based on partial KWBS data set as the following: 140 mm for
larch and swampy sparse growth forest, 110 mm in dwarf cedar and alder shrubs of tundra belt,
and about 70 mm for rocky talus.

The average values of evaporation from snow in mm per day are determined from
measurement data as follows: January-February – -0.04; March – +0,09; April – +0,40; May –
+0.74; September – +0,20; October – +0,01. Typical values of evaporation from snow for 1976-
1977 are presented at Fig. 11.

**4.4 Thaw/freeze depth**

The longest observation period is 33 continuous years (cryopedometer 17.5 located at the
forest with bushes, maximum thawing is 130 cm, 1964-1997). The deepest values of thawing
were observed in rocky talus landscape and can reach more than 240 cm. The shallowest values
of thawing range from 60 to 70 cm at swampy forest. Thawing of soils at the forest zone varies
in large ranges and depends on the location of the cryopedometer at a slope (Table 6, Fig. 12).

Lebedeva et al. (2014) reviewed the patterns of soil thaw/freeze processes and their impact on hydrological processes based on the analysis and modelling of the data at the cryopedometers in main landscapes of KWBS: rocky talus, mountain tundra with dwarf tree brush, moss-lichen cover and sparse-growth forest or larch forest.

**4.5 Streamflow**

Flow at KWBS begins in May, most of it occurs in summer. At the outlet of KWBS Kontaktovy creek at Nizhny 33, 24 and 20 %% of flow occurs in June, July and August respectively. For the summer period, rainfall floods are typical (Fig. 13).

Small brooks freeze completely in October. Surface flow stops at the channel of Kontaktovy creek at Nizhny gauge in November, but there is the evidence that the river valley talik located lower than the Kontaktovy-Nizhny gauge, the runoff exists till the beginning of snowmelt, which is evidenced by continuous drop of levels in hydrogeological wells (Glotov, 2002).

Annual runoff of the Kontaktovy stream basin with area 21.3 km$^2$ (average altitude 1070 m) is 281 mm for the period 1948-1997, it increases with the elevation and at the Morozova catchment (mean elevation 1370 m, basin area 0.63 km$^2$) reaches 453 mm (1969-1996). The flow from south-facing (Severny) and north-facing (Yuzhny) micro-watersheds with area of 0.38 and 0.27 km$^2$ are 227 and 193 mm for the period 1960-1997 respectively.

Maximum daily discharge was observed in August, 1979 and amounted to 7.6 m$^3$s$^{-1}$ (daily flow 30 mm) and 0.438 m$^3$s$^{-1}$ (60 mm) at the Kontaktovy – Nizhny and at the Morozova watersheds respectively.

**4.6 Changes of hydrometeorological elements in 50 years, 1948-1997**

The time series of flow characteristics and basic meteorological elements were evaluated for stationarity, in relation to presence of monotonic trends, with Mann-Kendall and Spearman rank-correlation tests, at the significance level of p $< 0.05$ (Mann 1945; Kendall 1975). If both tests proved a trend, a serial correlation coefficient was tested. With the serial correlation coefficient r $< 0.20$, the trend was considered reliable. In the case of r$\geq 0.20$, to eliminate autocorrelation in the input series «trend-free pre-whitening» procedure (TFPW), described by Yue (Yue et al. 2002), was carried out. «Whitened» time-series were repeatedly tested with Mann-Kendall non-parametric test. Trend value was estimated with Theil-Sen estimator (Sen 1968).

The annual air temperature at Nizhnyaya station increased by 1.1˚C, positive trends are observed in March and October accounting for the rise of temperature by 2.3 and 3.3 °C correspondingly. Annual sum of precipitation has grown by 74 mm (21%). Maximum annual daily precipitation has also increased by 8 mm, or 31%.

The analysis of monthly and annual flow (mm) for the Kontaktovy creek – Nizhny from 1948 to 1997 has revealed the changes of hydrological regime in those 50 years of runoff observations (Fig. 14). Positive trends of monthly flow are identified in May amounting to 29 mm, or 92%, as well as in October (5.7 mm, 166%) and November (0.35 mm, 252%). The annual flow trend increased by 67 mm, or 24%. These results confirm general situation of increasing low flow which is observed in Siberia (Tananaev et al., 2016) and North America (Yang et al., 2015; St. Jacques and Sauchyn, 2009).

**5. Water balance estimation**

The study of the water balance of watersheds is aimed at assessing the quantitative changes in its components, which makes it possible to study the main regularities in the runoff formation. In the northern regions, where climate change is more pronounced than in other parts of the world (Arctic Climate…, 2004) and standard hydrological network is shrinking (Shiklomanov et al, 2002), the assessment of the water balance and its future change is important.

The book Northern Research Basins Water Balance (2004) compiles the main results of water balance studies in the northern watersheds in last century such as Wolf Creek (Janowicz, et al., 2004), Kuparuk River (Lilly et al., 1998), Scotty Creek (Quinton et al., 2004), Nelka river (Vasilenko, 2004), including Kontaktovy Creek of KWBS (Zhuravin, 2004).

In this section the results of rough estimation of mean annual water balance for three micro-watersheds with area less than 1 km$^2$ and representative for main landscapes of studied territory (Severny, Yuzhny, Morozova) are presented and compared with the assessments made by other authors.

The estimation of water balance for the whole watershed of Kontaktovy cr. requires special analysis and does not lie in the scope of this paper; only the results of other authors are shortly summarized.

A general form of water balance equation (in mm) is used as the following:

$$SWE + P_{rain} + \Delta P_{corr} - ET - E_{snow} - R = \eta. \tag{1}$$

Here SWE is average value of snow water equivalent before spring freshet from snow surveys data. For Morozova watershed SWE is increased by 36 mm which is the average precipitation in May at ground rain gauge #42 (1400 m a.s.l.).

$P_{rain}$ is total sum of daily rainfall precipitation during warm period from the rain gauges located within studied watersheds. The data before 1969 was corrected for wetting losses according to (Manual for hydrometeorological stations …, 1969). For Severny and Yuzhny watersheds rainfall precipitation $P_{rain}$ is calculated as total sum of precipitation in May-August period and half of average precipitation in September accounting for air temperature transition from positive to negative which usually occurs in the mid of September at rain gauges #5 (880 m a.s.l.) and #20 (900 m a.s.l.) respectively. For Morozova watershed which is in average 300 m higher than Severny and Yuzhny ones, $P_{rain}$ consists of sum precipitation for the period from June to August estimated based on the data from daily precipitation data of rain gauge #38 (1200 m a.s.l.).

$\Delta P_{corr}$ is wind and evaporation correction of warm period rainfall precipitation calculated using the wind speed data from Nizhnyaya and Verknyaya stations based on the recommendations of Manual for hydrometeorological stations …, 1969).

ET, soil evapotranspiration, is calculated using average annual values for main landscapes of KWBS estimated by Semenova et al. (2013) and Lebedeva et al. (2017) with the account of their distribution across the studied watersheds.

Evaporation from snow $E_{snow}$ is assessed as the following:

$$E_{snow}=0.40*d_{Apr} + 0.74*d_{May} \tag{2}$$

where $d_{Apr}$ and $d_{May}$ are average numbers of days in April and May between the date of maximum SWE and its full melt; 0.40 and 0.74 are average values of snow evaporation in April and May estimated based on observed data.

R is observed runoff; and $\eta$ is an error term.

Possible members of water balance equation such as the changes in surface storage (lakes, wetlands, reservoirs, channels, etc.), subsurface storage of groundwater and the storage of unsaturated zone are estimated as zero and not accounted for long-term annual estimation.

Table 8 shows the distribution of water balance components for three small watersheds. All main components of water balance were assessed independently on the basis of data of direct observations. At two watersheds, the water balance discrepancy calculated as the difference between precipitation, runoff and total evaporation, is positive and varies from 43 to 57 mm which is about 11 and 14 % of calculated total precipitation. The water balance error at the Morozova watershed, which is completely formed of rocky talus, is negative and amounted to 68 mm or 14% from total precipitation. Though we did not use for calculations the data of solid precipitation, Sushansky (2002) assessed snow under-catch at Morozova watershed as 25-30 mm per year. Zhuravin (2004) mentioned that significant errors are possible at the sampling depth of snowpack profile in the areas covered with the Siberian dwarf-pine which is covered by snow during winter. He assessed the error of SWE estimation in such areas as 15% of the measured value. We would suggest that measuring SWE at rocky talus watershed where some areas are covered by boulders could cause the error of compared magnitude.

Estimated runoff coefficient amounts to 56, 51 and 95 % of precipitation for Severny, Yuzhny and Morozova watersheds respectively. Considering high runoff coefficient for rocky talus landscape, large proportion of the KWBS area (34%) covered by this type of underlying surface (Fig. 2B) and significant uncertainty of water balance estimation for Morozova Creek given the availability of observed data, correct assessment of water balance for larger areas seems rather complicated.

Table 8 presents the comparison of water balance calculations for three micro-watersheds of KWBS performed by different authors. While in this research SWE from snow surveys was taken as the estimate of winter precipitation, both – Lebedeva et al. (2017) and Zhuravin (2004) used observed precipitation data for assessing this component of water balance. The estimates of total precipitation vary due to different correction procedure applied (or not applied) by different authors. One may see that though all the authors used the same observed data on evaporation, its interpretation has provided the variation of results (Table 8). Also low closure error does not always confirm the correctness of estimation as, for example, Lebedeva et al. (2017) did not apply any bias correction to precipitation neither accessed the value of snow evaporation.

For the main KWBS watershed, the Kontakovy Creek (21.3 km$^2$), Zhuravin (2004) provided the following estimates of water balance for the period 1970-1985: precipitation – 405 mm, evaporation – 137 mm, runoff – 296 mm, discrepancy error - -28 mm (7%). Lebedeva et al. (2017) calculated the same values for 1949-1990 as 390, 281, 114 and -5 mm (-1%) respectively.

Presented results confirm that accurate numerical estimation of water balance elements even using available measurements is complicated (Kane and Yang, 2004) and subjective. Therefore it is important to make raw observational data available for scientific community as described in this paper.

**6. Data availability**

All data presented in this paper are available from the "PANGAEA. Data Publisher for Earth & Environmental Science" (see Makarieva et al., 2017, https://doi.pangaea.de/10.1594/PANGAEA.881731).

The directory includes 12 elements:

1. daily precipitation time series at 25 gauges within Kolyma Water-Balance Station (KWBS), 1948-1997;
2. daily runoff time series at ten gauges of KWBS, 1948-1997;
3. evaporation time series at 9 sites at KWBS, 1950-1997;
4. meteorological observations at three sites of KWBS, including the values of air temperature, water vapour pressure, vapour pressure deficit, atmospheric pressure, wind speed, low and total cloud amount, and surface temperature, 1948-1997;
5. monthly precipitation time series at 30 gauges within KWBS, 1948-1997;
6. precipitation (10 day sum) time series at 21 gauges within KWBS, 1962-1997;
7. precipitation (5 day sum) time series at 14 gauges within KWBS, 1966-1997;
8. snow survey line characteristics at KWBS, 1959-1997;
9. snow survey time series at different sites and landscapes within KWBS, 1950-1997;
10. soil temperature time series at the Nizhnyaya meteorological station at KWBS, 1974-1981;
11. thaw depth and snow height time series at different sites of KWBS, 1954-1997;
12. snow evaporation time series at two sites of KWBS, 1968-1992.

**7. The future of the KWBS**

[revised manuscript text omitted]

Based on the observation data annual water balance of three micro-watersheds (0.27 - 0.69 $km^2$) was estimated for the whole period of observations. Estimated runoff coefficients varied from 51-56 % at to 95 % in rocky talus. Assessment of water balance at larger scale is complicated due to significant uncertainty of water balance estimation for rocky talus which occupies about the third of the KWBS area.

Analysis of flow and meteorological data revealed general warming and the changes of water balance components in 1948-1997. The increase of annual air temperature amounted to 1.1 °C, annual precipitation has grown by 21%. Annual flow increased by 24%, positive trends were also determined in May (92%), October (166%), November (252%).

The dataset is important because it characterizes the natural settings, which, on the one hand, are nearly ungauged, and on the other hand, are representative for the vast mountainous territory of Eastern Siberia and North-East Russia. It is unique because it combines water balance, hydrological and permafrost data which allow for studying permafrost hydrology interaction processes within the range of all scientific issues, from models development to climate change impacts research.

**9. Author contribution**

O. Makarieva and N. Nesterova digitized and prepared the dataset for publication with assistance from L. Lebedeva and S. Sushansky. The data were collected in 1948-1997 by Hydrometeorological Service of USSR and Russia and published in Observation Reports (1959-1997).

**10. Acknowledgements**

Authors are grateful to two anonymous reviewers, Wolf-Dietrich Marchand and Pedro Restrepo for valuable input. Their comments allowed for better understanding of described data and inspire for the continuation of our fight for the restoration of the Kolyma Water-Balance station. We also thank David Post for English language correction.

[revised manuscript text omitted]

Table 8 Water balance of three micro-watersheds of KWBS (mm, %)

| Watershed | Severny | | | Yuzhny | | | Morozova | |
|---|---|---|---|---|---|---|---|---|
| Authors* | M | L | Z | M | L | Z | M | L |
| Period | 1958-1997 | 1959-1990 | 1970-1985 | 1960-1997 | 1960-1990 | 1970-1985 | 1969-1997 | 1969-1990 |
| SWE | 126 | - | - | 121 | - | - | 161+36 | - |
| $P_{rain}$ | 263 | - | - | 232 | - | - | 225 | - |
| $\Delta P_{corr}$ | 25 | - | 70 | 22 | - | 65 | 55 | - |
| $P_{total}$ | 375 | 357 | 399 | 375 | 332 | 346 | 477 | 451 |
| ET | 113 | 120 | - | 124 | 132 | - | 73 | 73 |
| $E_{snow}$ | 17 | 0 | - | 15 | 0 | - | 19 | 0 |
| $E_{total}$ | 130 | 120 | 139 | 139 | 132 | 147 | 92 | 73 |
| R | 227 | 236 | 217 | 193 | 199 | 190 | 453 | 448 |
| R (%) | 56 | 66 | 54 | 51 | 60 | 55 | 95 | 99 |
| η | 57 | 1 | 43 | 43 | 1 | 9 | -68 | -70 |
| η (%) | 14 | 0,3 | 11 | 11 | 0,2 | 3 | 14 | 16 |

*M – current research (Makarieva et al.); L – Lebedeva et al. (2017); Z – Zhuravin (2004).

[Figure]

Fig. 1 The view of the Kolyma Water Balance Station, A – August 2016 (the photo by O. Makarieva), B – historical photo from Sushansky (1989)

[Figure]

Fig. 2 Scheme of the Kolyma Water Balance Station (KWBS) indicating the location of observation sites

[Figure]

Fig. 3 A – Pluviometer, Tretyakov gauge, B – Kosarev gauge, C – remaining of ground gauge GR-28
                                (Sushansky, 1988, 1989)

[Figure]

Fig. 4 A, B – Snow survey at the Kontaktovy creek catchment; C – measurement of snow density, 1960
           (the photos from the KWBS archive, provided by S.I. Sushansky).

[Figure]

Fig. 5 A - Rykachev evaporimeter, B - weighing the GGI-500-30 evaporimeter (Sushansky, 1989)

[Figure]

Fig. 6 Runoff observations: A – runoff gauge at the Kontaktovy creek, 1979; B – runoff gauge at the
Dozhdemerny creek, 1959; C – runoff gauge at the Yuzhniy creek, 1960; D – runoff gauge at the
Vstrecha creek, 1953 (the photos from the KWBS archive, provided by S.I. Sushansky)

[Figure]

Fig. 7 Characteristics of discharge accuracy, 1984-1997. A – the percentage of extrapolation of stage
curves, B, C – the difference between measured and estimated maximum and minimum instant discharges
                              respectively

[Figure]

Fig. 8 Annual precipitation (mm) and air temperature (°C) at Nizhnyaya weather station, 1949-1996

[Figure]

Fig. 9 Mean monthly precipitation (mm) and air temperature (°C) at Nizhnyaya weather station, 1949-
                                 1996

[Figure]

Fig. 10 Snow water equivalent at the Kontaktovy creek and Morozova br.

[Figure]

Fig. 11 Snow evaporation (mm) during day and night period, 1976-1977

[Figure]

           Fig. 12 Depth of ground thawing at the different landscapes of KWBS

[Figure]

Fig. 13 Flow depth (mm) at the Kontaktovy creek – Nizhny (1102), Severny br. (1107) – south-facing
slope with cedar dwarf bush landscape and Morozova br. (1103) – rocky talus landscape at watershed
divides, 1970

[Figure]

Fig. 14 The trends of hydrometeorological elements, 1949-1997. A - annual precipitation at the
Nizhnyaya station; B - annual air temperatures at the Nizhnyaya station; C - annual flow at the
Kontaktovy creek – Nizhny; D - flow in October at the Kontaktovy creek – Nizhny

---

## Author Comment (AC4) · 10 Feb 2018

The authors thank Referee # 1 for his valuable comments. The responses are provided below. The changes are made in the text which is attached.

Major comment: The discussed data set is called "Kolyma water-balance station" implying that the water balance has been calculated here. However, only the different components of the water balance are presented in this manuscript, but not the balance. I know, it is very difficult to do that for such a location where winter conditions are extremely harsh. Nevertheless, I wonder whether or not the annual water balance has been calculated and suggest to include such results if available. Else you may explain why the term "water balance" is so prominent in the name of the station.

Response: Water-balance stations are a historical name of the network of the research watersheds that existed in former USSR. The overall goal of the water-balance stations network was detailed study of water balance components on slope and small scales in different environmental settings for the development of methods of hydrological forecast and flow characteristics assessments for engineering design. The KWBS was one of 26 water-balance stations of the USSR and the only located in the zone of continuous permafrost. The explanation is added to the text. Lines 48-55

Additional section 5 is added (lines 540-623). In this section the results of rough estimation of mean annual water balance for three micro-watersheds with area less than 1 km2 and representative for main landscapes of studied territory (Severny, Yuzhny, Morozova) are presented and compared with the assessments made by other authors. The estimation of water balance for the whole watershed of Kontaktovy cr. requires special analysis and does not lie in the scope of this paper; only the results obtained by other authors are shortly summarized.

Minor comments: - In the abstract you state that "the data are representative for the vast territory of the North-East of Russia". What is this claim based on? Response: Nasybulin (1976) showed that runoff characteristics at the KWBS are representative for the upper Kolyma River area. Taking into account the similarity of main landscape types across mountainous regions of North-East Russia to those found at the KWBS, the conclusion can be made that the KWBS hydrological conditions are actually representative for larger areas than described by Nasybulin (1976). These explanations are given in the text (lines 93-97)

"the data are representative for the vast territory of the North-East of Russia" was changed to "the data are representative for the mountainous territory of the North-East of Russia" in the abstract (line 18)

The reference is added to the reference list Nasybulin, P.S. (1976) The representativity of runoff characteristics at the Kolyma Water Balance Station for the upper Kolyma area. Natural resources of the USSR North-East. Vladivostok, AN DVIS IBPS, 32-41 (in Russian)

You mention different types of evaporimeters (Rykachev, Gorshenin, GGI-500), some of them used also for snow evaporation measurements. Could you briefly explain how these evaporimeters differ from each other. Response: Evaporimeter GGI-500-50 is a standard device for the soil evaporation measurements in Russia and former USSR. It consists of two cylindrical vessels, one inside the other, and a water-collecting vessel. The bottom of the inner cylinder has openings; the core sample is placed in it. The quantity of water evaporated is determined from the difference in weight of the sample as measured over two successive observation periods. Rykachev evaporimeter was used for the soil evaporation measurements in 1950s. It consists of a sealed square rectangular box with a core sample. The box was placed inside another box installed in the ground. Since the inner box is sealed the device doesn't account for infiltrated water. We could not find any information about Gorshenin evaporimeter. We continue searching. The clarification is given in the text (lines 335-346)

Also, when you measure snow evaporation, is this representative for snow on the ground? I assume (and know from our own longterm observations) that snow evaporation from trees (interception) is more relevant than from the ground. Can you comment on that? – Response: This measurement accounts for the snow evaporation on the ground only. In the conditions of KWBS with the larch as the main tree type, intercepted snow was only temporary phenomena because of cyclonic activity in January and February. Wind during the cyclones blows away snow from all trees except dwarf cedar that is under snow for the most part of the winter. The clarification is given in the text (lines 368-32)

I guess that most readers of this journal don't know what a Danilin cryopedometer is. I understand that it measures the thaw depth of the active layer, but how exactly? – Response: Danilin cryopedometer (frost tubes) was designed by Danilin (Snyder et al., 1971). Cryopedometer consists of a rubber tube 1 cm in external diameter and calibrated to an accuracy of 1 cm. The tube is filled with distilled water, closed at both ends and lowered into a casing (an ebonite pipe) installed in a borehole in the soil. In order to measure the depth of freezing, the rubber tube is taken from the casing and the lower end of the ice column in the tube was determined (Lebedeva et al, 2014). The explanation is introduced in the text (lines 387-391)

Lebedeva L., Semenova O., Vinogradova T. (2014) Simulation of Active Layer Dynamics, Upper Kolyma, Russia, using the Hydrograph Hydrological Model // Permafrost and Periglac. Process. 25 (4): 270–280 DOI: 10.1002/ppp.1821 Snyder F, Sokolov A, Szesztay K. 1971. Flood Studies: An International Guide for Collection and Processing of Data. Unesco: Paris; 52pp.

Some of the figures are just too small. For example, the map of Fig.2 should be enlarged to become readable. Also the very nice photos in Fig. 6 showing the hard work of measuring snow would deserve a larger format. Response: Figures are corrected.

Different types of rain gauges and shields are mentioned in section 3.2 (Nipher, Tretyakov, Kosarev and pit rain gauge), and a reference is made to Fig. 5. But which one of these types are actually shown in Fig. 5? Please clarify in the figure caption. The Figure 3 presents the photos of different types of precipitation gauges and the caption is clarified (lines 975-977)

And, if possible, could you shortly explain how these specific types differ from each other. Response: More details about precipitation data and its correction is provided, as well the description of different precipitation gauges and their use at KWBS. Lines 269-315

- line 260: "correspond to" instead of "account for" - line 275: "amount to" instead of

"account for" – line 309: add "the" after "published in" – line 415: remove "could" – Table 2: you may include the lower caption into the table

Response: corrections were made according to the comments

Please also note the supplement to this comment:
https://www.earth-syst-sci-data-discuss.net/essd-2017-125/essd-2017-125-AC4-supplement.pdf

**Supplement:**

[revised manuscript text omitted]

Though in the last several decades and more recently many research watersheds were established in Arctic zone of the USA and Canada, to the best of our knowledge, the first systematic cold-region hydrology observations in North America began not earlier than the 1960s. Such, the Caribou-Poker Creeks Research Watershed was established only in 1969 (Hinzman et al., 2002), 20 years later than KWBS.

One may mention numerous scientific catchments in Alaska – Fish Creek (Pacific Northwest…, 2014), Toolik station (Hoobbie et al., 2003), Tanana River (Yarie et al., 1998), Kuparuk River (Arp & Stuefer, 2017, Kane & Hinzman, 2009), Imnavait River (Walker & Walker, 1996), Putuligayuk River (Kane & Hinzman, 2009), as well as Arctic monitoring programs (NPR-Hydrology (NPR-A Hydrology…, 2018), Arctic Observatory Network (Arctic Observatory Network…, 2018).

The studies at research watersheds of Canada are integrated into scientific programs and accompanied by data analysis and models development and applications For example, the Changing Cold Regions Network project (Changing Cold…, 2018) includes field studies on 14 watersheds and the use of two Canadian models CHRM and CLASS. The Improving Processes & Parameterization for Prediction in Cold Regions Hydrology (IP3) project (Improving

Processes…, 2018) combined 10 research watersheds and four hydrological models – CHRM (Pomeroy et al., 2007), CLASS (Verseghy, 1991), MESH (Pietroniro et al., 2007), GEM (Yeh et al., 2002).

Although there are large mountainous areas in other cold regions of the world, the combination of extremely severe climate (mean annual air temperature reaches -11.3°C) and continuous permafrost creates unique conditions at KWBS which are not presented at any other research watershed of the world.

Nasybulin (1976) showed that hydrological regime at the KWBS is representative for the whole Upper Kolyma Plateau. Taking into account the similarity of main landscape types across mountainous regions of North-East Russia to those found at the KWBS, the conclusion can be made that hydrological conditions at KWBS are actually representative for vaster ungauged areas, than described by Nasybulin (1976).

Sufficiently long time series of observations which were continuously conducted by uniform methods and covered pre-warming period are of high importance for the studies of climate change impact on hydrology in the Arctic.

[revised manuscript text omitted]

The list of used devices and the accuracy of observations for each meteorological element is presented in the description files of the database.

**3.2 Precipitation**

In total, the precipitation was observed at 47 gauges within KWBS territory during different periods. Continuous daily all-year around precipitation data is available for the period 1948-1997 for the gauge (#12) at meteorological station Nizhnyaya and for the gauge (#54) at meteorological station Kulu for 1981-1997. Four gauges have the data of daily totals during warm season for the period for more than 30 years and another 18 gauges for different shorter periods. Usually the start of daily observations at those gauges was initiated by the beginning of snowmelt period and lasted until the end of September. Monthly sums of precipitation were measured at 30 gauges, 10-days and 5-days sums – at 21 and 18 gauges respectively.

In 1948 precipitation gauge stations for measuring daily precipitation were equipped with the Nipher-shielded and Tretyakov-shielded precipitation gauges (Fig. 3A). In 1948-1958 the observations were carried out with both devices in parallel, after 1959 only Tretyakov gauges were used. Tretyakov precipitation gauges were also used for measurements precipitation totals in 5 and 10 days periods.

The other types of precipitation gauges applied at the KWBS are the Kosarev and ground rain gauge (GR-28) (Fig. 3B, 3C). GR-28 gauge with receiving area 500 cm$^2$ was installed into the special box several cm above the ground. GR-28 were usually installed on the 1$^{st}$ of June and dismantled on the 1$^{st}$ of September and used for rain measurements over the longer period, typically one month. Only those GR-28 which were installed at the soil evaporation plots measured precipitation every day. The Kosarev precipitation gauges were used for monthly precipitation measurements from the 1$^{st}$ of October to the 1$^{st}$ of May. Different precipitation gauges are shown at Fig. 5.

In 1960-1963 there was an attempt to register precipitation with automatic radio-precipitation gauges, but due to improper performance of the devices those observations were stopped.

In 1988, precipitation observations at the KWBS were carried out with 36 precipitation and rain gauges, distributed relatively evenly throughout the area and altitudinal zones. Average density of the precipitation network at that time accounted for 1.6 units per 1 km$^2$.

For the period 1948-1968 precipitation data was published in Observation Reports as it was without any correction. Starting from 1969, all daily, 5-days and 10-days totals data from

Tretyakov rain gauges have been corrected for wetting losses according to Manual (1969). The correction value for precipitation event varied from 0.0 to 0.2 mm depending on the amount of observed precipitation and weather conditions. In average annual value of wetting losses correction did not exceed 5 % of total amount of precipitation, though in some years it could reach up to 9-10 %. In 1948-1983 monthly sums data obtained from GR-28 and Kosarev gauges were published without any correction. In 1984 wetting losses correction was introduced to GR-28 observations as well. In the database the precipitation data is presented in original form without any changes.

The analysis of water balance, climate change impact on river runoff or hydrological modelling requires accurate and reliable precipitation data. Arctic and mountainous regions are characterized by high bias of precipitation measurements because of significant amount of snowfall precipitation (WMO Report #67, 1998). Monthly estimates of this bias often vary from 5% to 40%. Biases are larger in winter than in summer largely due to the deleterious effect of the wind on snowfall (Groisman et al., 2015). Three main methods of winter precipitation bias correction are suggested for the Tretyakov gauge which was the main type of precipitation gauge at KWBS. They are the WMO methodology (Yang and Goodison et al., 1995), Northern European countries method (Forland et al., 2000) and the approach developed by Golubev (WMO Report #67, 1998). The basis of all three methods for correcting measured precipitation is the dependence of the aerodynamic coefficient on wind speed, air temperature, precipitation type and wind protection.

In described database each precipitation gauge (if it was available) has the description of its location, altitude, slope exposure, vegetation type. Additional characteristic is the degree of protection characterized by five types of Schwer (1976) classification (Ia, Ib – protected; IIa, IIb – half-protected; III – open; IV – shore station). The database also contains the series of daily wind speed for three meteorological stations which combined with the information on location gauges can be used as a proxy for introducing bias corrections.

**3.3 Snow surveys**

Snow cover observations were started in 1950 and initially conducted at two catchments, at two meteorological plots and four typical squares. In 1959-1960 the number of catchments with snow surveys reached five. Up to 1971, snow surveys were conducted once per month starting in November and finishing in May at small catchments (the Severny, Yuzhny, Dogdemerny and Vstrecha) and once before spring snowmelt at the Kontaktovy – Nizhny. Since 1972, the observations were reduced to one survey per year (usually at the end of April) for all watersheds. Table 2 shows the number of snow routes, their total length and number of measurement points, including their distribution among different landscapes of the catchments (Fig. 4). Snow depth was measured every 10 m, snow density – every 100 m at most of the watersheds, and 5 and 50 m respectively at the Morozova brook watershed.

Based on the data about measured snow height and snow weight with the account for landscape and elevation distribution average SWE for individual watersheds and landscapes was calculated and published in the Observation Reports.

Average depth of snow cover is presented with accuracy of 1 cm, density and SWE – 0.01 g cm$^{-3}$ and 1 mm respectively.

**3.4 Soil evaporation**

Three types of evaporimeters were used at the KWBS.

Evaporimeter GGI-500-50 (later modified to GGI-500-30) is a standard device for the soil evaporation measurements in Russia and former USSR (Fig. 5B). It consists of two cylindrical vessels, one inside the other, and a water-collecting vessel. The bottom of the inner cylinder has openings; the core sample is placed in it. The quantity of water evaporated is determined from the difference in weight of the sample as measured over two successive observation periods.

Rykachev evaporimeter was used for the soil evaporation measurements in 1950s. It consists of a sealed square rectangular box with a core sample (Fig. 5A). The box was placed inside another box installed in the ground. Since the inner box was sealed the device did not allow for water infiltration (Chebotarev, 1939).

The description of Gorshenin evaporimeter was not found.

[revised manuscript text omitted]

At the micro-watersheds of Morozova and Yuzhny brooks runoff was measured by means of a V-notch weir, at Severny brook – with a flow measuring flume.

In the database, mean daily values of streamflow are presented.

Originally daily discharges were published in Observation Reports in $l\,s^{-1}$ with accuracy of three significant figures, but not more accurately than $0.01\,l\,s^{-1}$ for runoff gauges equipped with weir or flume. For gauges with a natural channel for discharges more than $1000\,l\,s^{-1}$ the rounding to three significant figures was performed, for discharges less than $100\,l\,s^{-1}$ – to two significant figures, but not more precise than $1\,l\,s^{-1}$.

Small discharges which are less than $0.05\,l\,s^{-1}$ for the gauges equipped with hydrometric facilities and less than $0.5\,l\,s^{-1}$ for larger watersheds gauges were published in Observation Reports as 0.00 and 0, respectively. The periods with no runoff because of drying and freezing were marked with special symbols.

In the database, water discharges are converted to $m^3\,s^{-1}$, the number of significant figures was preserved but the values 0.00 and 0, as well as special symbols for freeze and dry periods are indicated as 0.

In 1984-1997 the information on the accuracy of discharge data was published for several runoff gauges In Observation Reports. It included the percentage of stage curve extrapolation in both directions which was published for one or several runoff periods per year depending on how many stage curves were applied. Also information about measured and estimated instant maximum and minimum discharges was available for the same period. Fig. 7 shows the boxplots of these characteristics for 7 runoff gauges for the period 1984-1997.

**4. Results**

**4.1 Meteorological variables and precipitation**

The climate of the study area is severely continental with harsh long winters and short but warm summers. Average annual temperature at the Nizhnyaya meteorological plot during 1949-1996 is -11.3 °C. Mean monthly temperature in January was -33.6 °C, in July +13.2 °C (Fig. 8-9). The absolute minimum daily temperature of -53.0 °C was registered in 1982 and the absolute maximum daily temperature was +22.8 °C (1988). The period of negative air temperatures lasts from October to April, freeze-free period is, on average, 130 days long.

Air temperature inversions are observed at the KWBS. In December air temperature gradient reaches +2.0, in May it accounts for -0.5°C per 100 m of elevation respectively.

The average air humidity at the Nizhnyaya station is 3.6 mb, reaching its maximum and minimum values of 9.8 and 0.4 mb in July and December respectively.

Total cloudiness at the Nizhnyaya station has average annual value of 7.0 and does not change considerably through the year. Its minimum and maximum values are 5.9 and 8.0 points in March and July. Lower cloudiness dynamic is more significant, its mean monthly values changes from 0.5 to 4.7 in March to July with average value of 2.2 points.

443  Mean wind velocity is more than twice higher at Verkhnyaya station (1220 m a.s.l.) in
444 comparison with Nizhnyaya station (850 m a.s.l.) and amount to 3.0 and 1.3 m s$^{-1}$ accordingly.
445 Average monthly values changes from 0.83 in December to 1.70 in May at Nizhnaya, and from
446 2.7 in November, February to 3.4 in May. Maximum daily wind speed amounted to 16 and 36 m
447 s$^{-1}$ at Nizhnyaya and Verkhnyaya.
448  Precipitation at Nizhnyaya meteorological plot from 1969 (the year when wetting losses
449 were introduced) to 1997 varied from 229 (1991) to 474 (1990) mm per year with mean value of
450 362 mm. After introducing wetting losses correction to the period 1949-1968 and computing
451 average mean amount from 1949 to 1997, its value decreased to 351 mm. Maximum and
452 minimum monthly amount of precipitation at Nizhnyaya station was observed in July and March
453 and correspond to 72 and 8 mm respectively for the whole period of observations 1949-1997
454 (Fig. 8-9).
455  Maximum daily amount of precipitation at Nizhnyaya station was observed in June 1968
456 reaching 48.1 mm. In average for the period of 50 years this statistic amounted to 26 mm.
458  **4.2 Snow cover**
459  Stable snow cover at KWBS in average is formed in the first weeks of October, and melts
460 in the third week of May (1949-1996). The KWBS area is characterized by an increase in the
461 thickness of the snow cover due to the absence of thaws during the whole snow season. In the
462 open treeless and watershed divide areas, the redistribution of snow pack due to wind blow is
463 observed.
464  Average for the watershed mean, maximum and minimum snow water equivalent (SWE)
465 before spring freshet estimated based on snow survey results at the Kontaktovy – Nizhny amount
466 to 121, 213 (1985) and 59 (1964) mm respectively in the 1960-1997 period. In general, rocky
467 talus and tundra bush landscape are characterized by lower SWE due to wind blowing. Much
468 snow is accumulated in the forest landscape. However, at the Morozova brook watershed which
469 is fully covered by rocky talus landscape, mean SWE before snowmelt was estimated as 161 mm
470 with the maximum value of 298 mm observed in 1985 reaching in average 0.99 m snow height
471 (Table 3, Table 4, Fig. 10).
473  **4.3 Soil and snow evaporation**
474  The highest values of soil evaporation during the summer period were observed at the
475 larch forest (site 9) and reached 136 mm. At a similar landscape (site 1), this value is lower, at
476 119 mm, which indicates the influence of local factors. The lowest values of soil evaporation are
477 104 mm at the plot located at dwarf cedar tree bush (site 7). In July, soil evaporation values
478 range from 33 to 40 mm, depending on the landscape. In September, the contribution of
479 evaporation decreases to 14-24 mm (Table 5).
480  Average values of annual soil evaporation were previously estimated by Semenova et al.
481 (2013) and Lebedeva et al. (2017) based on partial KWBS data set as the following: 140 mm for
482 larch and swampy sparse growth forest, 110 mm in dwarf cedar and alder shrubs of tundra belt,
483 and about 70 mm for rocky talus.
484  The average values of evaporation from snow in mm per day are determined from
485 measurement data as follows: January-February – -0.04; March – +0,09; April – +0,40; May –
486 +0.74; September – +0,20; October – +0,01. Typical values of evaporation from snow for 1976-
487 1977 are presented at Fig. 11.
489  **4.4 Thaw/freeze depth**
490  The longest observation period is 33 continuous years (cryopedometer 17.5 located at the
491 forest with bushes, maximum thawing is 130 cm, 1964-1997). The deepest values of thawing
492 were observed in rocky talus landscape and can reach more than 240 cm. The shallowest values
493 of thawing range from 60 to 70 cm at swampy forest. Thawing of soils at the forest zone varies
494 in large ranges and depends on the location of the cryopedometer at a slope (Table 6, Fig. 12).

Lebedeva et al. (2014) reviewed the patterns of soil thaw/freeze processes and their impact on hydrological processes based on the analysis and modelling of the data at the cryopedometers in main landscapes of KWBS: rocky talus, mountain tundra with dwarf tree brush, moss-lichen cover and sparse-growth forest or larch forest.

**4.5 Streamflow**

Flow at KWBS begins in May, most of it occurs in summer. At the outlet of KWBS Kontaktovy creek at Nizhny 33, 24 and 20 %% of flow occurs in June, July and August respectively. For the summer period, rainfall floods are typical (Fig. 13).

Small brooks freeze completely in October. Surface flow stops at the channel of Kontaktovy creek at Nizhny gauge in November, but there is the evidence that the river valley talik located lower than the Kontaktovy-Nizhny gauge, the runoff exists till the beginning of snowmelt, which is evidenced by continuous drop of levels in hydrogeological wells (Glotov, 2002).

Annual runoff of the Kontaktovy stream basin with area 21.3 km$^2$ (average altitude 1070 m) is 281 mm for the period 1948-1997, it increases with the elevation and at the Morozova catchment (mean elevation 1370 m, basin area 0.63 km$^2$) reaches 453 mm (1969-1996). The flow from south-facing (Severny) and north-facing (Yuzhny) micro-watersheds with area of 0.38 and 0.27 km$^2$ are 227 and 193 mm for the period 1960-1997 respectively.

Maximum daily discharge was observed in August, 1979 and amounted to 7.6 m$^3$s$^{-1}$ (daily flow 30 mm) and 0.438 m$^3$s$^{-1}$ (60 mm) at the Kontaktovy – Nizhny and at the Morozova watersheds respectively.

**4.6 Changes of hydrometeorological elements in 50 years, 1948-1997**

The time series of flow characteristics and basic meteorological elements were evaluated for stationarity, in relation to presence of monotonic trends, with Mann-Kendall and Spearman rank-correlation tests, at the significance level of $p < 0.05$ (Mann 1945; Kendall 1975). If both tests proved a trend, a serial correlation coefficient was tested. With the serial correlation coefficient $r < 0.20$, the trend was considered reliable. In the case of $r \geq 0.20$, to eliminate autocorrelation in the input series «trend-free pre-whitening» procedure (TFPW), described by Yue (Yue et al. 2002), was carried out. «Whitened» time-series were repeatedly tested with Mann-Kendall non-parametric test. Trend value was estimated with Theil-Sen estimator (Sen 1968).

The annual air temperature at Nizhnyaya station increased by 1.1˚C, positive trends are observed in March and October accounting for the rise of temperature by 2.3 and 3.3 °C correspondingly. Annual sum of precipitation has grown by 74 mm (21%). Maximum annual daily precipitation has also increased by 8 mm, or 31%.

The analysis of monthly and annual flow (mm) for the Kontaktovy creek – Nizhny from 1948 to 1997 has revealed the changes of hydrological regime in those 50 years of runoff observations (Fig. 14). Positive trends of monthly flow are identified in May amounting to 29 mm, or 92%, as well as in October (5.7 mm, 166%) and November (0.35 mm, 252%). The annual flow trend increased by 67 mm, or 24%. These results confirm general situation of increasing low flow which is observed in Siberia (Tananaev et al., 2016) and North America (Yang et al., 2015; St. Jacques and Sauchyn, 2009).

**5. Water balance estimation**

The study of the water balance of watersheds is aimed at assessing the quantitative changes in its components, which makes it possible to study the main regularities in the runoff formation. In the northern regions, where climate change is more pronounced than in other parts of the world (Arctic Climate…, 2004) and standard hydrological network is shrinking (Shiklomanov et al, 2002), the assessment of the water balance and its future change is important.

The book Northern Research Basins Water Balance (2004) compiles the main results of water balance studies in the northern watersheds in last century such as Wolf Creek (Janowicz, et al., 2004), Kuparuk River (Lilly et al., 1998), Scotty Creek (Quinton et al., 2004), Nelka river (Vasilenko, 2004), including Kontaktovy Creek of KWBS (Zhuravin, 2004).

In this section the results of rough estimation of mean annual water balance for three micro-watersheds with area less than 1 km$^2$ and representative for main landscapes of studied territory (Severny, Yuzhny, Morozova) are presented and compared with the assessments made by other authors.

The estimation of water balance for the whole watershed of Kontaktovy cr. requires special analysis and does not lie in the scope of this paper; only the results of other authors are shortly summarized.

A general form of water balance equation (in mm) is used as the following:

$$SWE + P_{rain} + \Delta P_{corr} - ET - E_{snow} - R = \eta. \tag{1}$$

Here SWE is average value of snow water equivalent before spring freshet from snow surveys data. For Morozova watershed SWE is increased by 36 mm which is the average precipitation in May at ground rain gauge #42 (1400 m a.s.l.).

$P_{rain}$ is total sum of daily rainfall precipitation during warm period from the rain gauges located within studied watersheds. The data before 1969 was corrected for wetting losses according to (Manual for hydrometeorological stations …, 1969). For Severny and Yuzhny watersheds rainfall precipitation $P_{rain}$ is calculated as total sum of precipitation in May-August period and half of average precipitation in September accounting for air temperature transition from positive to negative which usually occurs in the mid of September at rain gauges #5 (880 m a.s.l.) and #20 (900 m a.s.l.) respectively. For Morozova watershed which is in average 300 m higher than Severny and Yuzhny ones, $P_{rain}$ consists of sum precipitation for the period from June to August estimated based on the data from daily precipitation data of rain gauge #38 (1200 m a.s.l.).

$\Delta P_{corr}$ is wind and evaporation correction of warm period rainfall precipitation calculated using the wind speed data from Nizhnyaya and Verknyaya stations based on the recommendations of Manual for hydrometeorological stations …, 1969).

ET, soil evapotranspiration, is calculated using average annual values for main landscapes of KWBS estimated by Semenova et al. (2013) and Lebedeva et al. (2017) with the account of their distribution across the studied watersheds.

Evaporation from snow $E_{snow}$ is assessed as the following:

$$E_{snow}=0.40*d_{Apr} + 0.74*d_{May} \tag{2}$$

where $d_{Apr}$ and $d_{May}$ are average numbers of days in April and May between the date of maximum SWE and its full melt; 0.40 and 0.74 are average values of snow evaporation in April and May estimated based on observed data.

R is observed runoff; and $\eta$ is an error term.

Possible members of water balance equation such as the changes in surface storage (lakes, wetlands, reservoirs, channels, etc.), subsurface storage of groundwater and the storage of unsaturated zone are estimated as zero and not accounted for long-term annual estimation.

Table 8 shows the distribution of water balance components for three small watersheds. All main components of water balance were assessed independently on the basis of data of direct observations. At two watersheds, the water balance discrepancy calculated as the difference between precipitation, runoff and total evaporation, is positive and varies from 43 to 57 mm which is about 11 and 14 % of calculated total precipitation. The water balance error at the Morozova watershed, which is completely formed of rocky talus, is negative and amounted to 68 mm or 14% from total precipitation. Though we did not use for calculations the data of solid precipitation, Sushansky (2002) assessed snow under-catch at Morozova watershed as 25-30 mm per year. Zhuravin (2004) mentioned that significant errors are possible at the sampling depth of snowpack profile in the areas covered with the Siberian dwarf-pine which is covered by snow during winter. He assessed the error of SWE estimation in such areas as 15% of the measured value. We would suggest that measuring SWE at rocky talus watershed where some areas are covered by boulders could cause the error of compared magnitude.

Estimated runoff coefficient amounts to 56, 51 and 95 % of precipitation for Severny, Yuzhny and Morozova watersheds respectively. Considering high runoff coefficient for rocky talus landscape, large proportion of the KWBS area (34%) covered by this type of underlying surface (Fig. 2B) and significant uncertainty of water balance estimation for Morozova Creek given the availability of observed data, correct assessment of water balance for larger areas seems rather complicated.

Table 8 presents the comparison of water balance calculations for three micro-watersheds of KWBS performed by different authors. While in this research SWE from snow surveys was taken as the estimate of winter precipitation, both – Lebedeva et al. (2017) and Zhuravin (2004) used observed precipitation data for assessing this component of water balance. The estimates of total precipitation vary due to different correction procedure applied (or not applied) by different authors. One may see that though all the authors used the same observed data on evaporation, its interpretation has provided the variation of results (Table 8). Also low closure error does not always confirm the correctness of estimation as, for example, Lebedeva et al. (2017) did not apply any bias correction to precipitation neither accessed the value of snow evaporation.

For the main KWBS watershed, the Kontakovy Creek (21.3 km$^2$), Zhuravin (2004) provided the following estimates of water balance for the period 1970-1985: precipitation – 405 mm, evaporation – 137 mm, runoff – 296 mm, discrepancy error - -28 mm (7%). Lebedeva et al. (2017) calculated the same values for 1949-1990 as 390, 281, 114 and -5 mm (-1%) respectively.

Presented results confirm that accurate numerical estimation of water balance elements even using available measurements is complicated (Kane and Yang, 2004) and subjective. Therefore it is important to make raw observational data available for scientific community as described in this paper.

**6. Data availability**

All data presented in this paper are available from the "PANGAEA. Data Publisher for Earth & Environmental Science" (see Makarieva et al., 2017, https://doi.pangaea.de/10.1594/PANGAEA.881731).

The directory includes 12 elements:
1. daily precipitation time series at 25 gauges within Kolyma Water-Balance Station (KWBS), 1948-1997;
2. daily runoff time series at ten gauges of KWBS, 1948-1997;
3. evaporation time series at 9 sites at KWBS, 1950-1997;
4. meteorological observations at three sites of KWBS, including the values of air temperature, water vapour pressure, vapour pressure deficit, atmospheric pressure, wind speed, low and total cloud amount, and surface temperature, 1948-1997;
5. monthly precipitation time series at 30 gauges within KWBS, 1948-1997;
6. precipitation (10 day sum) time series at 21 gauges within KWBS, 1962-1997;
7. precipitation (5 day sum) time series at 14 gauges within KWBS, 1966-1997;
8. snow survey line characteristics at KWBS, 1959-1997;
9. snow survey time series at different sites and landscapes within KWBS, 1950-1997;
10. soil temperature time series at the Nizhnyaya meteorological station at KWBS, 1974-1981;
11. thaw depth and snow height time series at different sites of KWBS, 1954-1997;
12. snow evaporation time series at two sites of KWBS, 1968-1992.

**7. The future of the KWBS**

[revised manuscript text omitted]

Based on the observation data annual water balance of three micro-watersheds (0.27 - 0.69 km$^2$) was estimated for the whole period of observations. Estimated runoff coefficients varied from 51-56 % at to 95 % in rocky talus. Assessment of water balance at larger scale is complicated due to significant uncertainty of water balance estimation for rocky talus which occupies about the third of the KWBS area.

Analysis of flow and meteorological data revealed general warming and the changes of water balance components in 1948-1997. The increase of annual air temperature amounted to 1.1 °C, annual precipitation has grown by 21%. Annual flow increased by 24%, positive trends were also determined in May (92%), October (166%), November (252%).

The dataset is important because it characterizes the natural settings, which, on the one hand, are nearly ungauged, and on the other hand, are representative for the vast mountainous territory of Eastern Siberia and North-East Russia. It is unique because it combines water balance, hydrological and permafrost data which allow for studying permafrost hydrology interaction processes within the range of all scientific issues, from models development to climate change impacts research.

**9. Author contribution**

O. Makarieva and N. Nesterova digitized and prepared the dataset for publication with assistance from L. Lebedeva and S. Sushansky. The data were collected in 1948-1997 by Hydrometeorological Service of USSR and Russia and published in Observation Reports (1959-1997).

**10. Acknowledgements**

Authors are grateful to two anonymous reviewers, Wolf-Dietrich Marchand and Pedro Restrepo for valuable input. Their comments allowed for better understanding of described data and inspire for the continuation of our fight for the restoration of the Kolyma Water-Balance station. We also thank David Post for English language correction.

[revised manuscript text omitted]

Table 8 Water balance of three micro-watersheds of KWBS (mm, %)

| Watershed | Severny | | | Yuzhny | | | Morozova | |
|---|---|---|---|---|---|---|---|---|
| Authors* | M | L | Z | M | L | Z | M | L |
| Period | 1958-1997 | 1959-1990 | 1970-1985 | 1960-1997 | 1960-1990 | 1970-1985 | 1969-1997 | 1969-1990 |
| SWE | 126 | - | - | 121 | - | - | 161+36 | - |
| $P_{rain}$ | 263 | - | - | 232 | - | - | 225 | - |
| $\Delta P_{corr}$ | 25 | - | 70 | 22 | - | 65 | 55 | - |
| $P_{total}$ | 375 | 357 | 399 | 375 | 332 | 346 | 477 | 451 |
| ET | 113 | 120 | - | 124 | 132 | - | 73 | 73 |
| $E_{snow}$ | 17 | 0 | - | 15 | 0 | - | 19 | 0 |
| $E_{total}$ | 130 | 120 | 139 | 139 | 132 | 147 | 92 | 73 |
| R | 227 | 236 | 217 | 193 | 199 | 190 | 453 | 448 |
| R (%) | 56 | 66 | 54 | 51 | 60 | 55 | 95 | 99 |
| η | 57 | 1 | 43 | 43 | 1 | 9 | -68 | -70 |
| η (%) | 14 | 0,3 | 11 | 11 | 0,2 | 3 | 14 | 16 |

*M – current research (Makarieva et al.); L – Lebedeva et al. (2017); Z – Zhuravin (2004).

[Figure]

Fig. 1 The view of the Kolyma Water Balance Station, A – August 2016 (the photo by O. Makarieva), B – historical photo from Sushansky (1989)

[Figure]

Fig. 2 Scheme of the Kolyma Water Balance Station (KWBS) indicating the location of observation sites

[Figure]

Fig. 3 A – Pluviometer, Tretyakov gauge, B – Kosarev gauge, C – remaining of ground gauge GR-28
(Sushansky, 1988, 1989)

[Figure]

Fig. 4 A, B – Snow survey at the Kontaktovy creek catchment; C – measurement of snow density, 1960
            (the photos from the KWBS archive, provided by S.I. Sushansky).

[Figure]

Fig. 5 A - Rykachev evaporimeter, B - weighing the GGI-500-30 evaporimeter (Sushansky, 1989)

[Figure]

Fig. 6 Runoff observations: A – runoff gauge at the Kontaktovy creek, 1979; B – runoff gauge at the
Dozhdemerny creek, 1959; C – runoff gauge at the Yuzhniy creek, 1960; D – runoff gauge at the
Vstrecha creek, 1953 (the photos from the KWBS archive, provided by S.I. Sushansky)

[Figure]

Fig. 7 Characteristics of discharge accuracy, 1984-1997. A – the percentage of extrapolation of stage
curves, B, C – the difference between measured and estimated maximum and minimum instant discharges
                               respectively

[Figure]

Fig. 8 Annual precipitation (mm) and air temperature (°C) at Nizhnyaya weather station, 1949-1996

[Figure]

Fig. 9 Mean monthly precipitation (mm) and air temperature (°C) at Nizhnyaya weather station, 1949-
1996

[Figure]

Fig. 10 Snow water equivalent at the Kontaktovy creek and Morozova br.

[Figure]

  Fig. 11 Snow evaporation (mm) during day and night period, 1976-1977

[Figure]

  Fig. 12 Depth of ground thawing at the different landscapes of KWBS

[Figure]

Fig. 13 Flow depth (mm) at the Kontaktovy creek – Nizhny (1102), Severny br. (1107) – south-facing
slope with cedar dwarf bush landscape and Morozova br. (1103) – rocky talus landscape at watershed
divides, 1970

[Figure]

Fig. 14 The trends of hydrometeorological elements, 1949-1997. A - annual precipitation at the
Nizhnyaya station; B - annual air temperatures at the Nizhnyaya station; C - annual flow at the
Kontaktovy creek – Nizhny; D - flow in October at the Kontaktovy creek – Nizhny

---

## Author Comment (AC5) · 10 Feb 2018

Referee # 2 turned out to be the most tough and requiring in his comments. It was the pleasure to look for the responses which really helped us better understand and even more appreciate the data. The responses and description of made changes is given below. Corrected manuscript is attached.

The manuscript is well written and structured, however the authors should consider doing some structural edits according to suggestions below. Also, the title of the manuscript is misleading the reader since no water balance is presented for the study site. The different components of the water balance is presented, but no suggestions on how to set up the WB is given. I would recommend to change the title in order to better describe what is included in the manuscript.

Response: Water-balance stations are a historical name of the network of the research watersheds that existed in former USSR. The overall goal of the water-balance stations network was detailed study of water balance components on slope and small scales in different environmental settings for the development of methods of hydrological forecast and flow characteristics assessments for engineering design. The KWBS was one of 26 water-balance stations of the USSR and the only located in the zone of continuous permafrost. The explanation is added to the text. Lines 48-55 Additional section 5 is added (lines 540-623). In this section the results of rough estimation of mean annual water balance for three micro-watersheds with area less than 1 km2 and representative for main landscapes of studied territory (Severny, Yuzhny, Morozova) are presented and compared with the assessments made by other authors. The estimation of water balance for the whole watershed of Kontaktovy cr. requires special analysis and does not lie in the scope of this paper; only the results obtained by other authors are shortly summarized.

Specific comments: 1. Introduction: I recommend to go through the already published data sets in ESSD related to hydrological data in permafrost and arctic areas. It would be nice to get a more thorough picture of available data and how the data in the present manuscript complement already published hydrometeorological data from the arctic regions. Response: The changes to Introduction are made accordingly (lines 69-91)

2. Site description: The permafrost conditions is described. How about taliks in the area? Taliks have great impact on the interaction between permafrost and hydrological flows, describe shortly the presence of taliks in the areas and where they are found (under lakes or rivers) and what type of talik that is most common (open, close, through)

Response: No talik data is presented in the paper, so very short description of talik processes at the studied watershed is given at lines 153-159 "Along the whole length of the Kontaktovy Creek, channel taliks can be found. They go all the way through the layer of alluvial sediments and their depth may reach 15 m in the cross section of the Nizhny hydrological gauge (Mikhaylov, 2013) and 5 m on the flood plain (Glotov, 2002). In summer, the talik forms a single hydraulic system with waters of active layer and the creek channel. In winter it freezes only partially. In the talik located below Kontaktovy-Nizhny gauge, flow exists till the beginning of snowmelt, which is evidenced by continuous drop of levels in hydrogeological wells (Glotov, 2002)."

3. Data description: The data description and main results are given in the same section. I would recommend the authors to separate the technical description of equipment, installation techniques, measured time periods etc from result presentation of the collected data. A new chapter 4 presenting the main results for each parameter should facilitate for the reader. Inter- and intra annual patterns in the data should be presented in the results section and not in section 3 as it is presented in the present version of the manuscript. Response: We divided data description and main results into two different sections.

4. A table early in chapter 3 summarizing the measured parameters including details of measurement period, periods of data gaps and used equipment and/or methods for evaluation of data would give a better overview of the presented data, reference to observation points in the map in Figure 2 could also be listed in the table. Response: The details of measurements such as the periods, gaps, etc. are presented in the database additional files. We do not think it would be appropriate to present this rather long piece of information in the paper. Also the database contains detailed figures with all observation points and their references. In the paper we present just general figure with all observational points without their references to give the idea how dense the observational network was at the KWBS.

5. There is no or very little information about uncertainties and accuracy for the equipment used in the investigations. If information is available (given that the measurements were performed long time ago and technical descriptions of used equipment can be hard to find) a complementary section about uncertainties would rise the quality of the manuscript. Response: In the description of the data and measurement equipment the accuracy of measurements was specified where it was possible. For streamflow observations we analyzed and described the accuracy of data for the period 1984-1997 which was available in Observation Reports for several gauges in terms of 1) percentage of extrapolation of stage curve, 2) difference between measured and estimated instant maximum and minimum discharges. Additional Fig 7 with the boxplots of those characteristics was added.

6. Precipitation data: No details are given about the correction of precipitation data. I guess the data presented are uncorrected for wind and adhesion losses. Given that much of the precipitation fall as snow, the under-catch might be high and the errors due to this have to be discussed. Motivate why data is not corrected and provide the reader with necessary information about the location of the precipitation bucket/meteorological station to a proper correction can be made. The under-catch in wind exposed areas can be as high as 30-40% during the snowy season References to methods for correction and how this has been handled in other hydrometoerological studies should be given. Response: More details about precipitation data and its correction is provided, as well the description of different precipitation gauges and their use at KWBS. Short introduction into the problem of precipitation correction is given. Lines 269-315

7. The data in Pangaea: A complementary data set with maps in ArcGIS format would facilitate the use of the data-set in future studies. A base set of catchment geometries, land use, soil distribution, location of lakes and rivers, topography etc would make it much easier for data-users to set up proper hydrological models of the study site. Response: The maps presenting DEM, river network, location of observation gauges, main landscape distribution in ArcGis were added to the database.

8. Tables: The sites referred to in the tables are in general hard to find in the map in Figure 2. A clear coupling between site ID and the map must be given. The map, including the labels and legend, have to be enlarged. Figure 2 gives general view of the station. All maps with labels and references are attached in the database, now in ArcGis format as well. Figure 2 was enlarged.

9. Row 261: "Snow cover at KWBS is formed in the first weeks of October". . .based on data for which period? Give correct reference. Response: Corrected. Line 460-461 Stable snow cover at KWBS in average is formed in the first weeks of October, and melts in the third week of May (1949-1996).

10. Row 271: How is the SWE quantified? By weighing the snow or by calculation? Response: Based on the data about measured snow height and snow weight with the account for landscape and elevation distribution average SWE for individual watersheds and landscapes was calculated and published in the Observational Reports. Clarification is added at lines 328-330

Please also note the supplement to this comment:
https://www.earth-syst-sci-data-discuss.net/essd-2017-125/essd-2017-125-AC5-supplement.pdf

**Supplement:**

[revised manuscript text omitted]

Though in the last several decades and more recently many research watersheds were established in Arctic zone of the USA and Canada, to the best of our knowledge, the first systematic cold-region hydrology observations in North America began not earlier than the 1960s. Such, the Caribou-Poker Creeks Research Watershed was established only in 1969 (Hinzman et al., 2002), 20 years later than KWBS.

One may mention numerous scientific catchments in Alaska – Fish Creek (Pacific Northwest…, 2014), Toolik station (Hoobbie et al., 2003), Tanana River (Yarie et al., 1998), Kuparuk River (Arp & Stuefer, 2017, Kane & Hinzman, 2009), Imnavait River (Walker & Walker, 1996), Putuligayuk River (Kane & Hinzman, 2009), as well as Arctic monitoring programs (NPR-Hydrology (NPR-A Hydrology…, 2018), Arctic Observatory Network (Arctic Observatory Network…, 2018).

The studies at research watersheds of Canada are integrated into scientific programs and accompanied by data analysis and models development and applications For example, the Changing Cold Regions Network project (Changing Cold…, 2018) includes field studies on 14 watersheds and the use of two Canadian models CHRM and CLASS. The Improving Processes & Parameterization for Prediction in Cold Regions Hydrology (IP3) project (Improving

Processes…, 2018) combined 10 research watersheds and four hydrological models – CHRM (Pomeroy et al., 2007), CLASS (Verseghy, 1991), MESH (Pietroniro et al., 2007), GEM (Yeh et al., 2002).

Although there are large mountainous areas in other cold regions of the world, the combination of extremely severe climate (mean annual air temperature reaches -11.3°C) and continuous permafrost creates unique conditions at KWBS which are not presented at any other research watershed of the world.

Nasybulin (1976) showed that hydrological regime at the KWBS is representative for the whole Upper Kolyma Plateau. Taking into account the similarity of main landscape types across mountainous regions of North-East Russia to those found at the KWBS, the conclusion can be made that hydrological conditions at KWBS are actually representative for vaster ungauged areas, than described by Nasybulin (1976).

Sufficiently long time series of observations which were continuously conducted by uniform methods and covered pre-warming period are of high importance for the studies of climate change impact on hydrology in the Arctic.

[revised manuscript text omitted]

The list of used devices and the accuracy of observations for each meteorological element is presented in the description files of the database.

**3.2 Precipitation**

In total, the precipitation was observed at 47 gauges within KWBS territory during different periods. Continuous daily all-year around precipitation data is available for the period 1948-1997 for the gauge (#12) at meteorological station Nizhnyaya and for the gauge (#54) at meteorological station Kulu for 1981-1997. Four gauges have the data of daily totals during warm season for the period for more than 30 years and another 18 gauges for different shorter periods. Usually the start of daily observations at those gauges was initiated by the beginning of snowmelt period and lasted until the end of September. Monthly sums of precipitation were measured at 30 gauges, 10-days and 5-days sums – at 21 and 18 gauges respectively.

In 1948 precipitation gauge stations for measuring daily precipitation were equipped with the Nipher-shielded and Tretyakov-shielded precipitation gauges (Fig. 3A). In 1948-1958 the observations were carried out with both devices in parallel, after 1959 only Tretyakov gauges were used. Tretyakov precipitation gauges were also used for measurements precipitation totals in 5 and 10 days periods.

The other types of precipitation gauges applied at the KWBS are the Kosarev and ground rain gauge (GR-28) (Fig. 3B, 3C). GR-28 gauge with receiving area 500 cm$^2$ was installed into the special box several cm above the ground. GR-28 were usually installed on the 1$^{st}$ of June and dismantled on the 1$^{st}$ of September and used for rain measurements over the longer period, typically one month. Only those GR-28 which were installed at the soil evaporation plots measured precipitation every day. The Kosarev precipitation gauges were used for monthly precipitation measurements from the 1$^{st}$ of October to the 1$^{st}$ of May. Different precipitation gauges are shown at Fig. 5.

In 1960-1963 there was an attempt to register precipitation with automatic radio-precipitation gauges, but due to improper performance of the devices those observations were stopped.

In 1988, precipitation observations at the KWBS were carried out with 36 precipitation and rain gauges, distributed relatively evenly throughout the area and altitudinal zones. Average density of the precipitation network at that time accounted for 1.6 units per 1 km$^2$.

For the period 1948-1968 precipitation data was published in Observation Reports as it was without any correction. Starting from 1969, all daily, 5-days and 10-days totals data from

Tretyakov rain gauges have been corrected for wetting losses according to Manual (1969). The correction value for precipitation event varied from 0.0 to 0.2 mm depending on the amount of observed precipitation and weather conditions. In average annual value of wetting losses correction did not exceed 5 % of total amount of precipitation, though in some years it could reach up to 9-10 %. In 1948-1983 monthly sums data obtained from GR-28 and Kosarev gauges were published without any correction. In 1984 wetting losses correction was introduced to GR-28 observations as well. In the database the precipitation data is presented in original form without any changes.

The analysis of water balance, climate change impact on river runoff or hydrological modelling requires accurate and reliable precipitation data. Arctic and mountainous regions are characterized by high bias of precipitation measurements because of significant amount of snowfall precipitation (WMO Report #67, 1998). Monthly estimates of this bias often vary from 5% to 40%. Biases are larger in winter than in summer largely due to the deleterious effect of the wind on snowfall (Groisman et al., 2015). Three main methods of winter precipitation bias correction are suggested for the Tretyakov gauge which was the main type of precipitation gauge at KWBS. They are the WMO methodology (Yang and Goodison et al., 1995), Northern European countries method (Forland et al., 2000) and the approach developed by Golubev (WMO Report #67, 1998). The basis of all three methods for correcting measured precipitation is the dependence of the aerodynamic coefficient on wind speed, air temperature, precipitation type and wind protection.

In described database each precipitation gauge (if it was available) has the description of its location, altitude, slope exposure, vegetation type. Additional characteristic is the degree of protection characterized by five types of Schwer (1976) classification (Ia, Ib – protected; IIa, IIb – half-protected; III – open; IV – shore station). The database also contains the series of daily wind speed for three meteorological stations which combined with the information on location gauges can be used as a proxy for introducing bias corrections.

**3.3 Snow surveys**

Snow cover observations were started in 1950 and initially conducted at two catchments, at two meteorological plots and four typical squares. In 1959-1960 the number of catchments with snow surveys reached five. Up to 1971, snow surveys were conducted once per month starting in November and finishing in May at small catchments (the Severny, Yuzhny, Dogdemerny and Vstrecha) and once before spring snowmelt at the Kontaktovy – Nizhny. Since 1972, the observations were reduced to one survey per year (usually at the end of April) for all watersheds. Table 2 shows the number of snow routes, their total length and number of measurement points, including their distribution among different landscapes of the catchments (Fig. 4). Snow depth was measured every 10 m, snow density – every 100 m at most of the watersheds, and 5 and 50 m respectively at the Morozova brook watershed.

Based on the data about measured snow height and snow weight with the account for landscape and elevation distribution average SWE for individual watersheds and landscapes was calculated and published in the Observation Reports.

Average depth of snow cover is presented with accuracy of 1 cm, density and SWE – 0.01 g cm$^{-3}$ and 1 mm respectively.

**3.4 Soil evaporation**

Three types of evaporimeters were used at the KWBS.

Evaporimeter GGI-500-50 (later modified to GGI-500-30) is a standard device for the soil evaporation measurements in Russia and former USSR (Fig. 5B). It consists of two cylindrical vessels, one inside the other, and a water-collecting vessel. The bottom of the inner cylinder has openings; the core sample is placed in it. The quantity of water evaporated is determined from the difference in weight of the sample as measured over two successive observation periods.

Rykachev evaporimeter was used for the soil evaporation measurements in 1950s. It consists of a sealed square rectangular box with a core sample (Fig. 5A). The box was placed inside another box installed in the ground. Since the inner box was sealed the device did not allow for water infiltration (Chebotarev, 1939).

The description of Gorshenin evaporimeter was not found.

[revised manuscript text omitted]

At the micro-watersheds of Morozova and Yuzhny brooks runoff was measured by means of a V-notch weir, at Severny brook – with a flow measuring flume.

In the database, mean daily values of streamflow are presented.

Originally daily discharges were published in Observation Reports in $l\,s^{-1}$ with accuracy of three significant figures, but not more accurately than $0.01\,l\,s^{-1}$ for runoff gauges equipped with weir or flume. For gauges with a natural channel for discharges more than $1000\,l\,s^{-1}$ the rounding to three significant figures was performed, for discharges less than $100\,l\,s^{-1}$ – to two significant figures, but not more precise than $1\,l\,s^{-1}$.

Small discharges which are less than $0.05\,l\,s^{-1}$ for the gauges equipped with hydrometric facilities and less than $0.5\,l\,s^{-1}$ for larger watersheds gauges were published in Observation Reports as 0.00 and 0, respectively. The periods with no runoff because of drying and freezing were marked with special symbols.

In the database, water discharges are converted to $m^3\,s^{-1}$, the number of significant figures was preserved but the values 0.00 and 0, as well as special symbols for freeze and dry periods are indicated as 0.

In 1984-1997 the information on the accuracy of discharge data was published for several runoff gauges In Observation Reports. It included the percentage of stage curve extrapolation in both directions which was published for one or several runoff periods per year depending on how many stage curves were applied. Also information about measured and estimated instant maximum and minimum discharges was available for the same period. Fig. 7 shows the boxplots of these characteristics for 7 runoff gauges for the period 1984-1997.

**4. Results**

**4.1 Meteorological variables and precipitation**

The climate of the study area is severely continental with harsh long winters and short but warm summers. Average annual temperature at the Nizhnyaya meteorological plot during 1949-1996 is -11.3 °C. Mean monthly temperature in January was -33.6 °C, in July +13.2 °C (Fig. 8-9). The absolute minimum daily temperature of -53.0 °C was registered in 1982 and the absolute maximum daily temperature was +22.8 °C (1988). The period of negative air temperatures lasts from October to April, freeze-free period is, on average, 130 days long.

Air temperature inversions are observed at the KWBS. In December air temperature gradient reaches +2.0, in May it accounts for -0.5°C per 100 m of elevation respectively.

The average air humidity at the Nizhnyaya station is 3.6 mb, reaching its maximum and minimum values of 9.8 and 0.4 mb in July and December respectively.

Total cloudiness at the Nizhnyaya station has average annual value of 7.0 and does not change considerably through the year. Its minimum and maximum values are 5.9 and 8.0 points in March and July. Lower cloudiness dynamic is more significant, its mean monthly values changes from 0.5 to 4.7 in March to July with average value of 2.2 points.

443  Mean wind velocity is more than twice higher at Verkhnyaya station (1220 m a.s.l.) in
444 comparison with Nizhnyaya station (850 m a.s.l.) and amount to 3.0 and 1.3 m s$^{-1}$ accordingly.
445 Average monthly values changes from 0.83 in December to 1.70 in May at Nizhnaya, and from
446 2.7 in November, February to 3.4 in May. Maximum daily wind speed amounted to 16 and 36 m
447 s$^{-1}$ at Nizhnyaya and Verkhnyaya.
448  Precipitation at Nizhnyaya meteorological plot from 1969 (the year when wetting losses
449 were introduced) to 1997 varied from 229 (1991) to 474 (1990) mm per year with mean value of
450 362 mm. After introducing wetting losses correction to the period 1949-1968 and computing
451 average mean amount from 1949 to 1997, its value decreased to 351 mm. Maximum and
452 minimum monthly amount of precipitation at Nizhnyaya station was observed in July and March
453 and correspond to 72 and 8 mm respectively for the whole period of observations 1949-1997
454 (Fig. 8-9).
455  Maximum daily amount of precipitation at Nizhnyaya station was observed in June 1968
456 reaching 48.1 mm. In average for the period of 50 years this statistic amounted to 26 mm.
458  **4.2 Snow cover**
459  Stable snow cover at KWBS in average is formed in the first weeks of October, and melts
460 in the third week of May (1949-1996). The KWBS area is characterized by an increase in the
461 thickness of the snow cover due to the absence of thaws during the whole snow season. In the
462 open treeless and watershed divide areas, the redistribution of snow pack due to wind blow is
463 observed.
464  Average for the watershed mean, maximum and minimum snow water equivalent (SWE)
465 before spring freshet estimated based on snow survey results at the Kontaktovy – Nizhny amount
466 to 121, 213 (1985) and 59 (1964) mm respectively in the 1960-1997 period. In general, rocky
467 talus and tundra bush landscape are characterized by lower SWE due to wind blowing. Much
468 snow is accumulated in the forest landscape. However, at the Morozova brook watershed which
469 is fully covered by rocky talus landscape, mean SWE before snowmelt was estimated as 161 mm
470 with the maximum value of 298 mm observed in 1985 reaching in average 0.99 m snow height
471 (Table 3, Table 4, Fig. 10).
473  **4.3 Soil and snow evaporation**
474  The highest values of soil evaporation during the summer period were observed at the
475 larch forest (site 9) and reached 136 mm. At a similar landscape (site 1), this value is lower, at
476 119 mm, which indicates the influence of local factors. The lowest values of soil evaporation are
477 104 mm at the plot located at dwarf cedar tree bush (site 7). In July, soil evaporation values
478 range from 33 to 40 mm, depending on the landscape. In September, the contribution of
479 evaporation decreases to 14-24 mm (Table 5).
480  Average values of annual soil evaporation were previously estimated by Semenova et al.
481 (2013) and Lebedeva et al. (2017) based on partial KWBS data set as the following: 140 mm for
482 larch and swampy sparse growth forest, 110 mm in dwarf cedar and alder shrubs of tundra belt,
483 and about 70 mm for rocky talus.
484  The average values of evaporation from snow in mm per day are determined from
485 measurement data as follows: January-February – -0.04; March – +0,09; April – +0,40; May –
486 +0.74; September – +0,20; October – +0,01. Typical values of evaporation from snow for 1976-
487 1977 are presented at Fig. 11.
489  **4.4 Thaw/freeze depth**
490  The longest observation period is 33 continuous years (cryopedometer 17.5 located at the
491 forest with bushes, maximum thawing is 130 cm, 1964-1997). The deepest values of thawing
492 were observed in rocky talus landscape and can reach more than 240 cm. The shallowest values
493 of thawing range from 60 to 70 cm at swampy forest. Thawing of soils at the forest zone varies
494 in large ranges and depends on the location of the cryopedometer at a slope (Table 6, Fig. 12).

Lebedeva et al. (2014) reviewed the patterns of soil thaw/freeze processes and their impact on hydrological processes based on the analysis and modelling of the data at the cryopedometers in main landscapes of KWBS: rocky talus, mountain tundra with dwarf tree brush, moss-lichen cover and sparse-growth forest or larch forest.

**4.5 Streamflow**

Flow at KWBS begins in May, most of it occurs in summer. At the outlet of KWBS Kontaktovy creek at Nizhny 33, 24 and 20 %% of flow occurs in June, July and August respectively. For the summer period, rainfall floods are typical (Fig. 13).

Small brooks freeze completely in October. Surface flow stops at the channel of Kontaktovy creek at Nizhny gauge in November, but there is the evidence that the river valley talik located lower than the Kontaktovy-Nizhny gauge, the runoff exists till the beginning of snowmelt, which is evidenced by continuous drop of levels in hydrogeological wells (Glotov, 2002).

Annual runoff of the Kontaktovy stream basin with area 21.3 km$^2$ (average altitude 1070 m) is 281 mm for the period 1948-1997, it increases with the elevation and at the Morozova catchment (mean elevation 1370 m, basin area 0.63 km$^2$) reaches 453 mm (1969-1996). The flow from south-facing (Severny) and north-facing (Yuzhny) micro-watersheds with area of 0.38 and 0.27 km$^2$ are 227 and 193 mm for the period 1960-1997 respectively.

Maximum daily discharge was observed in August, 1979 and amounted to 7.6 m$^3$s$^{-1}$ (daily flow 30 mm) and 0.438 m$^3$s$^{-1}$ (60 mm) at the Kontaktovy – Nizhny and at the Morozova watersheds respectively.

**4.6 Changes of hydrometeorological elements in 50 years, 1948-1997**

The time series of flow characteristics and basic meteorological elements were evaluated for stationarity, in relation to presence of monotonic trends, with Mann-Kendall and Spearman rank-correlation tests, at the significance level of $p < 0.05$ (Mann 1945; Kendall 1975). If both tests proved a trend, a serial correlation coefficient was tested. With the serial correlation coefficient $r < 0.20$, the trend was considered reliable. In the case of $r \geq 0.20$, to eliminate autocorrelation in the input series «trend-free pre-whitening» procedure (TFPW), described by Yue (Yue et al. 2002), was carried out. «Whitened» time-series were repeatedly tested with Mann-Kendall non-parametric test. Trend value was estimated with Theil-Sen estimator (Sen 1968).

The annual air temperature at Nizhnyaya station increased by 1.1˚C, positive trends are observed in March and October accounting for the rise of temperature by 2.3 and 3.3 °C correspondingly. Annual sum of precipitation has grown by 74 mm (21%). Maximum annual daily precipitation has also increased by 8 mm, or 31%.

The analysis of monthly and annual flow (mm) for the Kontaktovy creek – Nizhny from 1948 to 1997 has revealed the changes of hydrological regime in those 50 years of runoff observations (Fig. 14). Positive trends of monthly flow are identified in May amounting to 29 mm, or 92%, as well as in October (5.7 mm, 166%) and November (0.35 mm, 252%). The annual flow trend increased by 67 mm, or 24%. These results confirm general situation of increasing low flow which is observed in Siberia (Tananaev et al., 2016) and North America (Yang et al., 2015; St. Jacques and Sauchyn, 2009).

**5. Water balance estimation**

The study of the water balance of watersheds is aimed at assessing the quantitative changes in its components, which makes it possible to study the main regularities in the runoff formation. In the northern regions, where climate change is more pronounced than in other parts of the world (Arctic Climate…, 2004) and standard hydrological network is shrinking (Shiklomanov et al, 2002), the assessment of the water balance and its future change is important.

The book Northern Research Basins Water Balance (2004) compiles the main results of water balance studies in the northern watersheds in last century such as Wolf Creek (Janowicz, et al., 2004), Kuparuk River (Lilly et al., 1998), Scotty Creek (Quinton et al., 2004), Nelka river (Vasilenko, 2004), including Kontaktovy Creek of KWBS (Zhuravin, 2004).

In this section the results of rough estimation of mean annual water balance for three micro-watersheds with area less than 1 km$^2$ and representative for main landscapes of studied territory (Severny, Yuzhny, Morozova) are presented and compared with the assessments made by other authors.

The estimation of water balance for the whole watershed of Kontaktovy cr. requires special analysis and does not lie in the scope of this paper; only the results of other authors are shortly summarized.

A general form of water balance equation (in mm) is used as the following:

$$SWE + P_{rain} + \Delta P_{corr} - ET - E_{snow} - R = \eta. \tag{1}$$

Here SWE is average value of snow water equivalent before spring freshet from snow surveys data. For Morozova watershed SWE is increased by 36 mm which is the average precipitation in May at ground rain gauge #42 (1400 m a.s.l.).

$P_{rain}$ is total sum of daily rainfall precipitation during warm period from the rain gauges located within studied watersheds. The data before 1969 was corrected for wetting losses according to (Manual for hydrometeorological stations …, 1969). For Severny and Yuzhny watersheds rainfall precipitation $P_{rain}$ is calculated as total sum of precipitation in May-August period and half of average precipitation in September accounting for air temperature transition from positive to negative which usually occurs in the mid of September at rain gauges #5 (880 m a.s.l.) and #20 (900 m a.s.l.) respectively. For Morozova watershed which is in average 300 m higher than Severny and Yuzhny ones, $P_{rain}$ consists of sum precipitation for the period from June to August estimated based on the data from daily precipitation data of rain gauge #38 (1200 m a.s.l.).

$\Delta P_{corr}$ is wind and evaporation correction of warm period rainfall precipitation calculated using the wind speed data from Nizhnyaya and Verknyaya stations based on the recommendations of Manual for hydrometeorological stations …, 1969).

ET, soil evapotranspiration, is calculated using average annual values for main landscapes of KWBS estimated by Semenova et al. (2013) and Lebedeva et al. (2017) with the account of their distribution across the studied watersheds.

Evaporation from snow $E_{snow}$ is assessed as the following:

$$E_{snow}=0.40*d_{Apr} + 0.74*d_{May} \tag{2}$$

where $d_{Apr}$ and $d_{May}$ are average numbers of days in April and May between the date of maximum SWE and its full melt; 0.40 and 0.74 are average values of snow evaporation in April and May estimated based on observed data.

R is observed runoff; and $\eta$ is an error term.

Possible members of water balance equation such as the changes in surface storage (lakes, wetlands, reservoirs, channels, etc.), subsurface storage of groundwater and the storage of unsaturated zone are estimated as zero and not accounted for long-term annual estimation.

Table 8 shows the distribution of water balance components for three small watersheds. All main components of water balance were assessed independently on the basis of data of direct observations. At two watersheds, the water balance discrepancy calculated as the difference between precipitation, runoff and total evaporation, is positive and varies from 43 to 57 mm which is about 11 and 14 % of calculated total precipitation. The water balance error at the Morozova watershed, which is completely formed of rocky talus, is negative and amounted to 68 mm or 14% from total precipitation. Though we did not use for calculations the data of solid precipitation, Sushansky (2002) assessed snow under-catch at Morozova watershed as 25-30 mm per year. Zhuravin (2004) mentioned that significant errors are possible at the sampling depth of snowpack profile in the areas covered with the Siberian dwarf-pine which is covered by snow during winter. He assessed the error of SWE estimation in such areas as 15% of the measured value. We would suggest that measuring SWE at rocky talus watershed where some areas are covered by boulders could cause the error of compared magnitude.

Estimated runoff coefficient amounts to 56, 51 and 95 % of precipitation for Severny, Yuzhny and Morozova watersheds respectively. Considering high runoff coefficient for rocky talus landscape, large proportion of the KWBS area (34%) covered by this type of underlying surface (Fig. 2B) and significant uncertainty of water balance estimation for Morozova Creek given the availability of observed data, correct assessment of water balance for larger areas seems rather complicated.

Table 8 presents the comparison of water balance calculations for three micro-watersheds of KWBS performed by different authors. While in this research SWE from snow surveys was taken as the estimate of winter precipitation, both – Lebedeva et al. (2017) and Zhuravin (2004) used observed precipitation data for assessing this component of water balance. The estimates of total precipitation vary due to different correction procedure applied (or not applied) by different authors. One may see that though all the authors used the same observed data on evaporation, its interpretation has provided the variation of results (Table 8). Also low closure error does not always confirm the correctness of estimation as, for example, Lebedeva et al. (2017) did not apply any bias correction to precipitation neither accessed the value of snow evaporation.

For the main KWBS watershed, the Kontakovy Creek (21.3 km$^2$), Zhuravin (2004) provided the following estimates of water balance for the period 1970-1985: precipitation – 405 mm, evaporation – 137 mm, runoff – 296 mm, discrepancy error - -28 mm (7%). Lebedeva et al. (2017) calculated the same values for 1949-1990 as 390, 281, 114 and -5 mm (-1%) respectively.

Presented results confirm that accurate numerical estimation of water balance elements even using available measurements is complicated (Kane and Yang, 2004) and subjective. Therefore it is important to make raw observational data available for scientific community as described in this paper.

**6. Data availability**

All data presented in this paper are available from the "PANGAEA. Data Publisher for Earth & Environmental Science" (see Makarieva et al., 2017, https://doi.pangaea.de/10.1594/PANGAEA.881731).

The directory includes 12 elements:
1. daily precipitation time series at 25 gauges within Kolyma Water-Balance Station (KWBS), 1948-1997;
2. daily runoff time series at ten gauges of KWBS, 1948-1997;
3. evaporation time series at 9 sites at KWBS, 1950-1997;
4. meteorological observations at three sites of KWBS, including the values of air temperature, water vapour pressure, vapour pressure deficit, atmospheric pressure, wind speed, low and total cloud amount, and surface temperature, 1948-1997;
5. monthly precipitation time series at 30 gauges within KWBS, 1948-1997;
6. precipitation (10 day sum) time series at 21 gauges within KWBS, 1962-1997;
7. precipitation (5 day sum) time series at 14 gauges within KWBS, 1966-1997;
8. snow survey line characteristics at KWBS, 1959-1997;
9. snow survey time series at different sites and landscapes within KWBS, 1950-1997;
10. soil temperature time series at the Nizhnyaya meteorological station at KWBS, 1974-1981;
11. thaw depth and snow height time series at different sites of KWBS, 1954-1997;
12. snow evaporation time series at two sites of KWBS, 1968-1992.

**7. The future of the KWBS**

[revised manuscript text omitted]

Based on the observation data annual water balance of three micro-watersheds (0.27 - 0.69 km$^2$) was estimated for the whole period of observations. Estimated runoff coefficients varied from 51-56 % at to 95 % in rocky talus. Assessment of water balance at larger scale is complicated due to significant uncertainty of water balance estimation for rocky talus which occupies about the third of the KWBS area.

Analysis of flow and meteorological data revealed general warming and the changes of water balance components in 1948-1997. The increase of annual air temperature amounted to 1.1 °C, annual precipitation has grown by 21%. Annual flow increased by 24%, positive trends were also determined in May (92%), October (166%), November (252%).

The dataset is important because it characterizes the natural settings, which, on the one hand, are nearly ungauged, and on the other hand, are representative for the vast mountainous territory of Eastern Siberia and North-East Russia. It is unique because it combines water balance, hydrological and permafrost data which allow for studying permafrost hydrology interaction processes within the range of all scientific issues, from models development to climate change impacts research.

**9. Author contribution**

O. Makarieva and N. Nesterova digitized and prepared the dataset for publication with assistance from L. Lebedeva and S. Sushansky. The data were collected in 1948-1997 by Hydrometeorological Service of USSR and Russia and published in Observation Reports (1959-1997).

**10. Acknowledgements**

Authors are grateful to two anonymous reviewers, Wolf-Dietrich Marchand and Pedro Restrepo for valuable input. Their comments allowed for better understanding of described data and inspire for the continuation of our fight for the restoration of the Kolyma Water-Balance station. We also thank David Post for English language correction.

[revised manuscript text omitted]

Table 8 Water balance of three micro-watersheds of KWBS (mm, %)

| Watershed | Severny | | | Yuzhny | | | Morozova | |
|---|---|---|---|---|---|---|---|---|
| Authors* | M | L | Z | M | L | Z | M | L |
| Period | 1958-1997 | 1959-1990 | 1970-1985 | 1960-1997 | 1960-1990 | 1970-1985 | 1969-1997 | 1969-1990 |
| SWE | 126 | - | - | 121 | - | - | 161+36 | - |
| $P_{rain}$ | 263 | - | - | 232 | - | - | 225 | - |
| $\Delta P_{corr}$ | 25 | - | 70 | 22 | - | 65 | 55 | - |
| $P_{total}$ | 375 | 357 | 399 | 375 | 332 | 346 | 477 | 451 |
| ET | 113 | 120 | - | 124 | 132 | - | 73 | 73 |
| $E_{snow}$ | 17 | 0 | - | 15 | 0 | - | 19 | 0 |
| $E_{total}$ | 130 | 120 | 139 | 139 | 132 | 147 | 92 | 73 |
| R | 227 | 236 | 217 | 193 | 199 | 190 | 453 | 448 |
| R (%) | 56 | 66 | 54 | 51 | 60 | 55 | 95 | 99 |
| η | 57 | 1 | 43 | 43 | 1 | 9 | -68 | -70 |
| η (%) | 14 | 0,3 | 11 | 11 | 0,2 | 3 | 14 | 16 |

*M – current research (Makarieva et al.); L – Lebedeva et al. (2017); Z – Zhuravin (2004).

[Figure]

Fig. 1 The view of the Kolyma Water Balance Station, A – August 2016 (the photo by O. Makarieva), B – historical photo from Sushansky (1989)

[Figure]

Fig. 2 Scheme of the Kolyma Water Balance Station (KWBS) indicating the location of observation sites

[Figure]

Fig. 3 A – Pluviometer, Tretyakov gauge, B – Kosarev gauge, C – remaining of ground gauge GR-28
(Sushansky, 1988, 1989)

[Figure]

Fig. 4 A, B – Snow survey at the Kontaktovy creek catchment; C – measurement of snow density, 1960
            (the photos from the KWBS archive, provided by S.I. Sushansky).

[Figure]

Fig. 5 A - Rykachev evaporimeter, B - weighing the GGI-500-30 evaporimeter (Sushansky, 1989)

[Figure]

Fig. 6 Runoff observations: A – runoff gauge at the Kontaktovy creek, 1979; B – runoff gauge at the
Dozhdemerny creek, 1959; C – runoff gauge at the Yuzhniy creek, 1960; D – runoff gauge at the
Vstrecha creek, 1953 (the photos from the KWBS archive, provided by S.I. Sushansky)

[Figure]

Fig. 7 Characteristics of discharge accuracy, 1984-1997. A – the percentage of extrapolation of stage
curves, B, C – the difference between measured and estimated maximum and minimum instant discharges
                               respectively

[Figure]

Fig. 8 Annual precipitation (mm) and air temperature (°C) at Nizhnyaya weather station, 1949-1996

[Figure]

Fig. 9 Mean monthly precipitation (mm) and air temperature (°C) at Nizhnyaya weather station, 1949-
1996

[Figure]

Fig. 10 Snow water equivalent at the Kontaktovy creek and Morozova br.

[Figure]

  Fig. 11 Snow evaporation (mm) during day and night period, 1976-1977

[Figure]

  Fig. 12 Depth of ground thawing at the different landscapes of KWBS

[Figure]

Fig. 13 Flow depth (mm) at the Kontaktovy creek – Nizhny (1102), Severny br. (1107) – south-facing
slope with cedar dwarf bush landscape and Morozova br. (1103) – rocky talus landscape at watershed
divides, 1970

[Figure]

Fig. 14 The trends of hydrometeorological elements, 1949-1997. A - annual precipitation at the
Nizhnyaya station; B - annual air temperatures at the Nizhnyaya station; C - annual flow at the
Kontaktovy creek – Nizhny; D - flow in October at the Kontaktovy creek – Nizhny